# Linking global terrestrial CO$_2$ fluxes and environmental drivers: inferences from the Orbiting Carbon Observatory-2 and terrestrial biospheric models

Zichong Chen[1], Junjie Liu[2], Daven K. Henze[3], Deborah N. Huntzinger[4], Kelley C. Wells[5], Stephen Sitch[6], Pierre Friedlingstein[7], Emilie Joetzjer[8], Vladislav Bastrikov[9], Daniel S. Goll[10], Vanessa Haverd[11], Atul K. Jain[12], Etsushi Kato[13], Sebastian Lienert[14], Danica L. Lombardozzi[15], Patrick C. McGuire[16], Joe R. Melton[17], Julia E. M. S. Nabel[18], Benjamin Poulter[19], Hanqin Tian[20], Andrew J. Wiltshire[21], Sönke Zaehle[22], and Scot M. Miller[1]

[1]Department of Environmental Health and Engineering, Johns Hopkins University, Baltimore, MD, USA
[2]Jet Propulsion Laboratory, California Institute of Technology, Pasadena, CA, USA
[3]Department of Mechanical Engineering, University of Colorado Boulder, Boulder, CO, USA
[4]School of Earth and Sustainability, Northern Arizona University, Flagstaff, AZ, USA
[5]Department of Soil, Water, and Climate, University of Minnesota-Twin Cities, St. Paul, MN, USA
[6]College of Life and Environmental Sciences, University of Exeter, Exeter, UK
[7]College of Engineering, Mathematics and Physical Sciences, University of Exeter, Exeter, UK
[8]Centre National de Recherche Meteorologique, Unite mixte de recherche 3589 Meteo-France/CNRS, 42 Avenue Gaspard Coriolis, 31100 Toulouse, France
[9]Laboratoire des Sciences du Climat et de l'Environnement, Institut Pierre-Simon Laplace, CEA-CNRS-UVSQ, CE Orme des Merisiers, 91191 Gif-sur-Yvette CEDEX, France
[10]Université Paris Saclay, CEA-CNRS-UVSQ, LSCE/IPSL, Gif sur Yvette, France
[11]CSIRO Oceans and Atmosphere, G.P.O. Box 1700, Canberra, ACT 2601, Australia
[12]Department of Atmospheric Sciences, University of Illinois, Urbana, IL, USA
[13]Institute of Applied Energy (IAE), Minato-ku, Tokyo 105-0003, Japan
[14]Climate and Environmental Physics, Physics Institute and Oeschger Centre for Climate Change Research, University of Bern, Bern, Switzerland
[15]National Center for Atmospheric Research, Climate and Global Dynamics, Terrestrial Sciences Section, Boulder, CO, USA
[16]Department of Meteorology, Department of Geography & Environmental Science, National Centre for Atmospheric Science, University of Reading, Reading, UK
[17]Climate Research Division, Environment and Climate Change Canada, Victoria, BC, Canada
[18]Max Planck Institute for Meteorology, Hamburg, Germany
[19]NASA Goddard Space Flight Center, Biospheric Sciences Laboratory, Greenbelt, MD, USA
[20]International Center for Climate and Global Change Research, School of Forestry and Wildlife Sciences, Auburn University, 602 Ducan Drive, Auburn, AL, USA
[21]Met Office Hadley Centre, FitzRoy Road, Exeter EX1 3PB, UK
[22]Max Planck Institute for Biogeochemistry, P.O. Box 600164, Hans-Knöll-Str. 10, 07745 Jena, Germany

**Correspondence:** Zichong Chen (zchen74@jhu.edu)

**Abstract.** Observations from the Orbiting Carbon Observatory 2 (OCO-2) satellite have been used to estimate $CO_2$ fluxes in many regions of the globe and provide new insight into the global carbon cycle. The objective of this study is to infer the relationships between patterns in OCO-2 observations and environmental drivers (e.g., temperature, precipitation) and therefore inform a process understanding of carbon fluxes using OCO-2. We use a multiple regression and inverse model, and the regression coefficients quantify the relationships between observations from OCO-2 and environmental driver datasets within individual years for 2015–2018 and within seven global biomes. We subsequently compare these inferences to the relationships estimated from 15 terrestrial biosphere models (TBMs) that participated in the TRENDY model inter-comparison. Using OCO-2, we are able to quantify only a limited number of relationships between patterns in atmospheric $CO_2$ observations and patterns in environmental driver datasets (i.e., 10 out of the 42 relationships examined). We further find that the ensemble of TBMs exhibits a large spread in the relationships with these key environmental driver datasets. The largest uncertainty in the models is in the relationship with precipitation, particularly in the tropics, with smaller uncertainties for temperature and photosynthetically-active radiation (PAR). Using observations from OCO-2, we find that precipitation is associated with increased $CO_2$ uptake in all tropical biomes, a result that agrees with half of the TBMs. By contrast, the relationships that we infer from OCO-2 for temperature and PAR are similar to the ensemble mean of the TBMs, though the results differ from many individual TBMs. These results point to the limitations of current space-based observations for inferring environmental relationships but also indicate the potential to help inform key relationships that are very uncertain in state-of-the-art TBMs.

## 1  Introduction

Over the past decade, the field of space-based $CO_2$ monitoring has undergone a rapid evolution. The sheer number of $CO_2$-observing satellites has greatly increased, including GOSAT/GOSAT-2 (Kuze et al., 2009; Nakajima et al., 2012), TanSat (Yang et al., 2018) and OCO-2/OCO-3 (Crisp, 2015; Eldering et al., 2019). This expanding observing system provides atmospheric $CO_2$ observations broadly across the globe, making it possible to estimate the distribution and magnitude of $CO_2$ fluxes in many regions that have sparse in situ surface atmospheric $CO_2$ monitoring (e.g., the tropics and the Southern Hemisphere). For example, the OCO-2 satellite, launched in July 2014, provides ∼65,000 observations per day that pass quality screening (Eldering et al., 2017); this dense, global set of OCO-2 observations, combined with inverse modeling techniques, has been used to constrain regional- and continental-scale $CO_2$ sources and sinks (e.g., Eldering et al., 2017; Liu et al., 2017; Crowell et al., 2019; Palmer et al., 2019; Byrne et al., 2020a).

Recent advances in OCO-2 retrievals from the NASA ACOS science team have led to widespread reductions in observation errors (e.g., O'Dell et al., 2018). Reducing the errors in satellite observations of $CO_2$ is critical for understanding $CO_2$ sources and sinks using inverse modeling, as even small biases in the observations can have an impact on the $CO_2$ flux estimate (e.g., Chevallier et al., 2007; Feng et al., 2016; Chevallier et al., 2014; Miller et al., 2018). For example, Miller et al. (2018) evaluated the extent to which OCO-2 retrievals can detect patterns in biospheric $CO_2$ fluxes and found that an early version of the OCO-2 retrievals (version 7) is only equipped to provide accurate flux constraints across very large continental or hemispheric regions; by contrast, in a follow-up paper, Miller and Michalak (2020) re-visited satellite capabilities in light of recently improved

OCO-2 retrievals, and the authors argued that new OCO-2 retrievals can be used to constrain $CO_2$ fluxes for more detailed regions (i.e., for seven global biomes).

A further challenge is to use these new global satellite datasets to evaluate and improve process-based estimates of the global carbon cycle provided by terrestrial biospheric models (TBMs). TBMs have become an integral tool for understanding regional- and global-scale carbon dynamics and for predicting future carbon cycling under changing climate. With that said, existing TBMs show large uncertainties in carbon flux estimates at multiple spatial and temporal scales – at regional and seasonal scales (e.g., Peng et al., 2014; King et al., 2015), at global and inter-annual scales (e.g., Piao et al., 2020), and in historical and future projections (e.g., Friedlingstein et al., 2006; Huntzinger et al., 2017).

One approach to inform TBM development is to estimate flux totals using atmospheric observations and compare those totals against TBMs – to inform the magnitude, seasonality, or spatial distribution of fluxes (e.g., King et al., 2015; Bastos et al., 2018). A more challenging approach is to estimate the relationships between $CO_2$ fluxes and environmental drivers using atmospheric observations and compare those relationships directly to the relationships in TBMs. We define the term "environmental drivers" as any meteorological variables or characteristics of the physical environment that can be modeled or measured and may correlate with net ecosystem exchange (NEE). Several studies have shown that these types of comparisons are feasible using in situ atmospheric observations (e.g., Dargaville et al., 2002; Forkel et al., 2016; Gourdji et al., 2008, 2012; Piao et al., 2013, 2017; Wang et al., 2014; Fang and Michalak, 2015; Shiga et al., 2018; Wang et al., 2020). Among other studies, Fang and Michalak (2015) used in situ atmospheric $CO_2$ observations across North America and an inverse model-ing framework to probe the relationships between NEE and environmental drivers; the authors compared these relationships directly to those inferred from several TBMs, and found that TBMs have reasonable skill in representing the relationship with shortwave radiation but show weak performance in describing relationships with other drivers like water availability. Similarly, Shiga et al. (2018) used tower-based atmospheric $CO_2$ observations to explore regional interannual variability (IAV) in NEE across North America, and found that TBMs disagree on the dominant regions responsible for IAV; this disagreement can be linked to differing sensitivities of $CO_2$ fluxes to environmental drivers within the TBMs. At even longer temporal scale, Wang et al. (2014) employed atmospheric $CO_2$ growth rate record from Mauna Loa, Hawaii, USA and the South Pole for five decades to explore the sensitivity of the global $CO_2$ growth rate to tropical temperature; the authors found that existing TBMs do not capture the observed sensitivity of the growth rate to tropical climatic variability, implying a limited ability of these TBMs in representing the impact of drought and warming on tropical carbon dynamics.

More recently, a handful of studies have shown that it is possible to tease out relationships between $CO_2$ fluxes and environ-mental drivers using global satellite observations of $CO_2$ (e.g., Liu et al., 2017; Byrne et al., 2020b). For example, Liu et al. (2017) used observations from OCO-2 to disentangle the environmental processes related to flux anomalies in tropical regions during the 2015-2016 El Niño. Byrne et al. (2020b) assimilated in situ and GOSAT observations of atmospheric $CO_2$ and an inverse model framework, and found contrasting environmental sensitivities of IAV in $CO_2$ fluxes between western and eastern temperate North America.

The goal of this study is to use atmospheric $CO_2$ observations from OCO-2 to quantify the relationships between spatiotem-poral patterns in $CO_2$ fluxes and patterns in environmental driver datasets. We conduct this analysis for years 2015 – 2018 and

focus on relationships that manifest across an individual year and individual biome. We specifically quantify the relationships using a top-down regression framework and a geostatistical inverse model (GIM). We then compare the relationships inferred using OCO-2 observations against those inferred from 15 state-of-the-art TBMs from the TRENDY model comparison project (v8, https://sites.exeter.ac.uk/trendy; see Table S1 for a full list of TBMs; Sitch et al., 2015; Friedlingstein et al., 2019). The primary objectives of this analysis are threefold: (1) evaluate what kinds of environmental relationships we can infer using current satellite observations from OCO-2, (2) assess where and when TBMs do and do not show consensus on the relationships between $CO_2$ fluxes and salient environmental drivers, and (3) compare the relationships inferred from OCO-2 against those inferred from TBMs with the goal of informing and improving TBM development.

## 2 Methods

### 2.1 Overview

We quantify the relationships between $CO_2$ observations from OCO-2 and environmental driver datasets for different regions of the globe using a top-down regression framework and a GIM. We cannot directly observe the relationships between $CO_2$ fluxes and environmental driver datasets. With that said, an overarching idea of this study is that these relationships manifest in atmospheric $CO_2$ observations, and we can quantify at least some of these relationships using observations from OCO-2 and a weighted, multiple regression. The coefficients estimated as part of the regression relate patterns in atmospheric $CO_2$ observations to patterns in the environmental driver datasets.

As part of this analysis, we also explore differences in the estimated environmental relationships (i.e., regression coefficients) among different years and different biomes. To this end, we estimate separate regression coefficients for each of seven different global biomes, and we estimate separate coefficients for each individual year of the study period (2015 – 2018). Hence, each coefficient estimated here represents the relationship between OCO-2 observations and an environmental driver dataset across an entire year and a global biome. Miller and Michalak (2020) explored when and where current OCO-2 observations can be use to detect variability in surface $CO_2$ fluxes, and the authors argue that, in most seasons, the satellite can be used to constrain fluxes from seven large biome-based regions. Hence the choice of the seven biomes used in this study (Fig. 1).

We first conduct this analysis using $CO_2$ observations from OCO-2. We then conduct a parallel analysis using the outputs of 15 terrestrial biosphere models (TBMs) from the TRENDY model inter-comparison project (v8). The goal of this step is to compare the environmental relationships (i.e., regression coefficients) that we infer from OCO-2 against the regression coefficients that we estimate from numerous state-of-the-art TBMs. We can then identify any similarities or differences between the TBMs and inferences using OCO-2 observations. We specifically analyze TRENDY model outputs for years 2015–2018, the same years as the OCO-2 analysis described above. To conduct this analysis, we generate synthetic OCO-2 observations using each of the 15 TBMs and using an atmospheric transport model. We then run the multiple regression on these synthetic observations. This setup mirrors that of Fang and Michalak (2015) and creates an apples-to-apples comparison between the TBMs and OCO-2 observations; in each case, we use atmospheric observations (either real or synthetic) and use the same set of equations to estimate the regression coefficients.

The multiple regression used in this study has the following mathematical form (e.g., Fang and Michalak, 2015):

$$\boldsymbol{z} = h(\mathbf{X}\boldsymbol{\beta} + \boldsymbol{\zeta}) + \boldsymbol{\epsilon} \tag{1}$$

where $\boldsymbol{z}$ ($n \times 1$) is a vector of real or synthetic $CO_2$ observations from OCO-2, $\mathbf{X}$ ($m \times p$) is a matrix of environmental driver datasets (described in Sect. 2.2), and $\boldsymbol{\beta}$ ($p \times 1$) are the regression coefficients that are estimated as part of the regression. Each column of $\mathbf{X}$ represents a different environmental driver dataset for a specific biome in a specific year. Note that we estimate all of the coefficients for the different environmental drivers and different biomes simultaneously in the regression model. In addition, $\boldsymbol{\zeta}$ ($m \times 1$) represents patterns in the fluxes that cannot be described by the environmental driver datasets, and these values are unknown. This component of the fluxes is also commonly referred to as the stochastic component and is discussed in Sect. 2.5. $h()$ is an atmospheric transport model (described later in this section) that relates surface $CO_2$ fluxes ($\mathbf{X}\boldsymbol{\beta} + \boldsymbol{\zeta}$) to the atmospheric $CO_2$ observations, and $\boldsymbol{\epsilon}$ ($n \times 1$) is a vector of errors in the OCO-2 observations and/or in the atmospheric model. The statistical properties of these errors are estimated before running the regression (described in Sect. 2.4).

Note that this framework assumes linear relationships between the environmental driver datasets and the OCO-2 observations. Numerous existing studies have used linear models to approximate relationships with environmental driver datasets. For example, studies have used linear models to compare the relationships between $CO_2$ fluxes and environmental driver datasets in TBMs (e.g., Huntzinger et al., 2011), to infer these relationships using eddy flux observations (e.g., Mueller et al., 2010; Yadav et al., 2010), and to infer relationships between atmospheric $CO_2$ observations and environmental driver datasets (e.g., Gourdji et al., 2012; Fang et al., 2014; Fang and Michalak, 2015; Piao et al., 2013, 2017; Rödenbeck et al., 2018).

The equations above require an atmospheric transport model ($h()$). We use the forward GEOS-Chem model (version v9-02; http://www.geos-chem.org) in this study, and we further use wind fields from the Modern-Era Retrospective Analysis for Research and Applications (MERRA-2) to drive atmospheric transport within GEOS-Chem (Gelaro et al., 2017). The GEOS-Chem simulations used here have a global spatial resolution of 4° latitude by 5° longitude and therefore are best able to capture broad, regional spatial patterns in atmospheric $CO_2$.

## 2.2 Environmental driver datasets

We estimate the relationships between OCO-2 observations (either real or synthetic) and environmental driver datasets drawn from commonly-used meteorological reanalysis. We specifically consider the following driver datasets as predictor variables in the multiple regression: 2-meter air temperature, precipitation, photosynthetically active radiation (PAR), downwelling short-wave radiation, and specific humidity.

We also include a non-linear function of 2-meter air temperature as an environmental driver dataset in the regression (refered to hereafter as scaled temperature; plotted in Fig. S1 and described in detail in Supplement Sect. S3). Numerous existing studies show that the relationship between temperature and photosynthesis has a different sign depending upon the temperature range; at sufficiently warm temperatures, an increase in temperature yields a decrease in photosynthesis (e.g., Baldocchi et al., 2017). The scaled temperature function considered here can account for those differences, and we find that this function yields a better model-data fit in the regression analysis than using temperature alone. The scaled temperature function used here is from the

Vegetation Photosynthesis and Respiration Model (VPRM) (Mahadevan et al., 2008) and describes the non-linear relationship between temperature and photosynthesis (Raich et al., 1991). The function is shaped like an upside-down parabola (shown in Fig. S1). Furthermore, this type of non-linear temperature function has been commonly used in existing TBMs (e.g., Heskel et al., 2016; Luus et al., 2017; Dayalu et al., 2018; Chen et al., 2019).

The environmental driver datasets described above are drawn from the Climatic Research Unit (CRU) and Japanese Reanalysis (JRA) meteorology product (CRUJRA; Harris, 2019). We use environmental driver data from CRUJRA because it is the same product used to generate the TRENDY model estimates. All flux outputs from TRENDY are provided at a monthly temporal resolution, so we input monthly meteorological variables from CRUJRA into the regression framework. Furthermore, we regrid the environmental driver datasets to a 4° latitude by 5° longitude spatial resolution before inputting these datasets into the regression. This spatial resolution matches the resolution of the atmospheric transport simulations used in this study (described in Sect. 2.1). The regression coefficients therefore quantify the relationships between OCO-2 observations and patterns in environmental driver datasets that manifest at this spatial and temporal resolution.

We subsequently re-run the regression analysis using environmental driver datasets drawn from a second meteorological product. Estimates of environmental driver data like temperature or precipitation can vary among meteorological models, and these differences among models are a source of uncertainty in the estimated regression coefficients. Hence, the use of a second meteorological product can at least partially account for these uncertainties. We specifically re-run the regression analysis using environmental driver datasets drawn from MERRA-2. We choose MERRA-2 because it is a commonly used, global reanalysis product from the NASA Global Modeling and Assimilation Office (GMAO). Furthermore, we use wind fields from MERRA-2 to drive all atmospheric transport model simulations in this study (described in Sect. 2.1), so the use of MERRA-2 for the environmental driver datasets in the regression creates consistency with the wind fields in the atmospheric model simulations that support the regression.

Note that we do not include any remote sensing indices (e.g., solar-induced chlorophyll fluorescence or leaf area index) in the present study. Rather, the focus of this study is to explore environmental drivers of $CO_2$ fluxes, not remote sensing proxies for $CO_2$ fluxes. Also note that we standardize (i.e., normalize) each of the environmental driver datasets within each biome and each year before running the regression, as has been done in several previous GIM studies (e.g., Gourdji et al., 2012; Fang and Michalak, 2015). This step means that all of the estimated regression coefficients ($\beta$) have the same units, are independent of the original units on the environmental driver data, and can be directly compared to one another.

## 2.3  Model selection

We use model selection to decide which environmental driver datasets to include in the analysis of the OCO-2 observations and in the analysis of each TBM using synthetic observations. Model selection ensures that the environmental driver datasets in the regression ($\mathbf{X}$) do not overfit the available OCO-2 data ($\mathbf{z}$). The inclusion of additional environmental driver datasets or columns in $\mathbf{X}$ will always improve the model-data fit in the regression, but the inclusion of too many driver datasets in $\mathbf{X}$ can overfit the regression to available OCO-2 data and result in unrealistic coefficients ($\beta$) (e.g., Zucchini, 2000). In addition, model selection indicates which relationships with environmental drivers we can confidently constrain and which we cannot

given current OCO-2 observations (e.g., Miller et al., 2018). In this study, we implement a type of model selection known as the Bayesian Information Criterion (BIC) (Schwarz et al., 1978), and various forms of the BIC have been implemented in numerous recent atmospheric inverse modeling studies (e.g., Gourdji et al., 2012; Miller et al., 2013; Fang and Michalak, 2015; Miller et al., 2018; Miller and Michalak, 2020). Using the BIC, we score different combinations of environmental driver datasets that could be included in $\mathbf{X}$ based on how well each combination helps reproduce either the real or synthetic OCO-2 observations ($\mathbf{z}$, Eq. 1). We specifically use an implementation of the BIC from Miller et al. (2018) and Miller and Michalak (2020) that is designed to be computationally efficient for very large satellite datasets. The BIC scores in this implementation are calculated using the following equation:

$$BIC = L + p \ln n^* \tag{2}$$

where $L$ is the log likelihood of a particular combination of environmental driver datasets (i.e., columns of $\mathbf{X}$), $p$ is the number of environmental driver datasets in a particular combination, and $n^*$ is the effective number of independent observations. This last variable accounts for the fact that not all atmospheric observations are independent, and the model-data residuals can exhibit spatially and temporally correlated errors (Miller et al., 2018). For all simulations here, we use an estimate of $n^*$ for the v9 OCO-2 observations from Miller and Michalak (2020). The first component of Eq. 2 ($L$) rewards combinations that are a better fit to the OCO-2 observations ($\mathbf{z}$), whereas the second component of Eq. 2 ($p \ln n^*$) penalizes models with a greater number of columns to prevent overfitting. The best combination of environmental drivers for $\mathbf{X}$ is the combination that receives the lowest BIC score. Miller et al. (2018) describes this implementation of the BIC in greater detail, including the specific setup and equations.

Note that we run model selection for the OCO-2 data and re-run model selection for each set of synthetic OCO-2 datasets generated using each TBM. As a result, we sometimes select different environmental driver datasets for the analysis using different TBMs. This setup parallels that of Huntzinger et al. (2011) and Fang and Michalak (2015). Furthermore, we use the same set of environmental driver datasets in each year of the study period (e.g., 2015–2018), a setup that parallels existing GIM studies that use multiple years of atmospheric observations (e.g., Shiga et al., 2018). We estimate different regression coefficients ($\boldsymbol{\beta}$) for each year of the study period, but the actual environmental driver datasets included in the regression does not change from one year to the next. An environmental driver dataset is either selected to be included for all years in a specific biome (based on the BIC scores), or it is not used in any year of the analysis.

## 2.4 Statistical model for estimating the coefficients ($\boldsymbol{\beta}$)

Once we have chosen a set of environmental driver datasets using model selection, we estimate the coefficients ($\boldsymbol{\beta}$) that relate the real or synthetic OCO-2 observations to these environmental datasets (e.g., Gourdji et al., 2012; Fang and Michalak, 2015):

$$\hat{\boldsymbol{\beta}} = (h(\mathbf{X})^T \boldsymbol{\Psi}^{-1} h(\mathbf{X}))^{-1} h(\mathbf{X})^T \boldsymbol{\Psi}^{-1} \mathbf{z} \tag{3}$$

where $\boldsymbol{\Psi}$ ($n \times n$) is a covariance matrix that describes model-data residuals (discussed at the end of this section). Furthermore, the uncertainties in these estimated coefficients can also be estimated using a linear equation (e.g., Gourdji et al., 2008; Fang

and Michalak, 2015):

$$\mathbf{V}_{\hat{\boldsymbol{\beta}}} = (h(\mathbf{X})^T \boldsymbol{\Psi}^{-1} h(\mathbf{X}))^{-1} \qquad (4)$$

where $\mathbf{V}_{\hat{\boldsymbol{\beta}}}$ is a $p \times p$ covariance matrix.

We test out two different formulations for the covariance matrix $\boldsymbol{\Psi}$ to evaluate the sensitivity of the results to the assumptions made about the covariance matrix parameters. In one set of simulations, we model $\boldsymbol{\Psi}$ as a diagonal matrix. The diagonal values characterize model-data errors ($\epsilon$), estimated for the version 9 retrievals from the recent OCO-2 model inter-comparison project (e.g., Crowell et al., 2019). The values have an average standard deviation of 0.98 ppm and range from 0.29 ppm to 4.8 ppm. In a second set of simulations, we use a more complex and more complete formulation of $\boldsymbol{\Psi}$: $\boldsymbol{\Psi} = h(h(\mathbf{Q})^T) + \mathbf{R}$

(e.g., Fang and Michalak, 2015), where $\mathbf{R}$ ($n \times n$) characterizes the model-data errors (described above), and $\mathbf{Q}$ ($m \times m$) is a covariance matrix that describes $\boldsymbol{\zeta}$ (the patterns in the fluxes that cannot be described by the environmental driver datasets). This formulation is more complete because it fully accounts for the residuals between $\boldsymbol{z}$ and $\mathbf{X}\boldsymbol{\beta}$. However, it is extremely computationally intensive to estimate the coefficients ($\boldsymbol{\beta}$) using this complex formulation of $\boldsymbol{\Psi}$. We cannot explicitly formulate this more complex version of $\boldsymbol{\Psi}$ due to its large size and the number of atmospheric model simulations ($h()$) that would be

required. As a result, we find the solution to Eq. 3 using this complex version of $\boldsymbol{\Psi}$ by iteratively minimizing the cost function for a geostatistical inverse model (GIM) (Sects. 2.5 and S1), a process that takes approximately two weeks for each year of model simulations in the setup used here.

We use both the simple and complex formulations of $\boldsymbol{\Psi}$ when analyzing the real OCO-2 observations. Both the simple and complex formulations of $\boldsymbol{\Psi}$ yield similar estimates for the coefficients $\boldsymbol{\beta}$, as discussed in the Results & Discussion (Sect. 3.2).

When analyzing the 15 TRENDY models, we only use the simple, diagonal formulation of $\boldsymbol{\Psi}$ – because of the prohibitive computational costs that would be required to run the more complex approach for all 15 TRENDY models.

Note that we estimate the values of $\mathbf{Q}$, the covariance matrix that describes $\boldsymbol{\zeta}$, using an approach known as restricted maximum likelihood estimation (e.g., Mueller et al., 2008; Gourdji et al., 2008, 2010, 2012). In the SI, we discuss the structure of $\mathbf{Q}$ in detail, describe RML, and compare the estimated parameters for $\mathbf{Q}$ against existing studies.

## 2.5    Statistical model for estimating $CO_2$ fluxes using OCO-2 observations

To complement the analysis described above, we take an additional step for the real OCO-2 observations of estimating $\boldsymbol{\zeta}$, patterns in the fluxes that cannot be described by the environmental driver datasets, also known as the stochastic component of the fluxes (Eq. 1). This step thereby creates a complete estimate of $CO_2$ fluxes using OCO-2 observations. This additional step accomplishes two goals. First, the fluxes in $\boldsymbol{\zeta}$ can reveal flux anomalies or patterns that are too complex to quantify using

a linear combination of environmental variables and/or can indicate the strengths and shortfalls of the regression. Second, by estimating all components of the $CO_2$ fluxes ($\mathbf{X}\boldsymbol{\beta}$ and $\boldsymbol{\zeta}$), we can better evaluate our inferences using OCO-2 against independent, ground-based observations of $CO_2$. This independent evaluation is important because OCO-2 observations and the atmospheric transport model (i.e., GEOS-Chem) can contain errors.

We generate a complete estimate of the $CO_2$ fluxes ($\mathbf{X}\boldsymbol{\beta} + \boldsymbol{\zeta}$) by minimizing the cost function for a GIM (e.g., Kitanidis, 1986; Michalak et al., 2004; Miller et al., 2020). We describe this process in detail in Supplemental Sect. S1. This process requires two covariance matrices ($\mathbf{R}$ and $\mathbf{Q}$), and we use the same parameters for these covariance matrices as described above in Sect. 2.4. Note that for the setup here, we estimate $\boldsymbol{\zeta}$ at a spatial resolution of $4°$ latitude by $5°$ longitude to match that of GEOS-Chem, and we estimate $\boldsymbol{\zeta}$ at a daily temporal resolution to better account for sub-monthly variability in $CO_2$ fluxes. Also note that minimizing the GIM cost function yields the same estimate for the coefficients ($\boldsymbol{\beta}$) as in Eq. 3, provided that the covariance matrices in the GIM cost function and in Eq. 3 are identical. The Supplemental Sect. S1 and Miller et al. (2020) describe the process of minimizing the GIM cost function in greater detail.

## 2.6 Analysis using real observations from OCO-2

For analysis using OCO-2 observations, we employ 10-second averages of the version 9 OCO-2 observations (e.g., Crowell et al., 2019) and include both land nadir- and land glint-mode retrievals. Recent retrieval updates have greatly reduced biases that previously existed between land nadir and land glint observations (O'Dell et al., 2018). Moreover, Miller and Michalak (2020) evaluated the impact of these updated OCO-2 retrievals on the terrestrial $CO_2$ flux constraint in different regions of the globe; the authors found that the inclusion of both land nadir and land glint retrievals yielded a stronger constraint on $CO_2$ fluxes relative to using only a single observation type.

We also include a column of $\mathbf{X}$ in all simulations using real OCO-2 observations to account for anthropogenic emissions, ocean fluxes, and biomass burning. This column includes anthropogenic emissions from the Open-Data Inventory for Anthropogenic Carbon Dioxide (ODIAC) (Oda et al., 2018), ocean fluxes from NASA Estimating the Circulation and Climate of the Ocean (ECCO) Darwin (Carroll et al., 2020), and biomass burning fluxes from the Global Fire Emissions Database (GFED) (Randerson et al., 2018). We estimate a single coefficient or scaling factor ($\boldsymbol{\beta}$) for this column. These fluxes are input into the regression at a $4°$ latitude by $5°$ longitude spatial resolution to match that of GEOS-Chem. The supplemental Sect. S2 contains greater discussion of these $CO_2$ sources.

## 2.7 Analysis using the TBMs

We compare the estimated coefficients ($\boldsymbol{\beta}$) from real OCO-2 observations against simulations using synthetic OCO-2 observations generated from 15 different TBMs in TRENDY (v8). We list out all of the individual models in the TRENDY comparison in Table S1. Model outputs from the TRENDY project were provided at a monthly time resolution, and the spatial resolution varies from one model to another (though many models have a native spatial resolution of either $0.5°$ latitude-longitude or $1°$ latitude-longitude). We specifically use TRENDY model outputs from scenario 3 simulations, in which all TBMs are forced with time-varying $CO_2$, climate, and land use.

We generate synthetic OCO-2 observations using each of these TBM flux estimates. To do so, we first regrid each of the TRENDY model estimates to a spatial resolution of $4°$ latitude by $5°$ longitude, the spatial resolution of the GEOS-Chem model. We then run the TRENDY model fluxes through the GEOS-Chem model for years $2015 - 2018$ and interpolate the model outputs to the times and locations of the OCO-2 observations.

## 3 Results & discussion

### 3.1 Results of model selection

The model selection results highlight the strengths and limitations of using current OCO-2 observations to estimate relation-ships with environmental driver datasets. We use model selection based on the BIC to determine a set of environmental driver datasets to include in the analysis using OCO-2 observations and using the TBMs. We only select 10 environmental driver datasets when we run model selection on the OCO-2 observations – both when we use environmental driver datasets from the CRUJRA and MERRA-2 products. We are generally able to identify at least one or two key environmental relationships in each biome using total column $CO_2$ observations (shown on the x-axis of Fig. 3). With that said, we are only able to quantify relationships with these few, salient environmental variables. More detailed environmental relationships within each biome are difficult to discern.

Note that we select a similar number of environmental driver datasets when using synthetic OCO-2 observations that are generated from each of the TBMs. We select anywhere between 8 and 13 environmental driver datasets (an average of 10 datasets) in the analysis using each of the TBMs. This result indicates consistency between the analysis using real OCO-2 observations and the analysis using synthetic OCO-2 observations generated using $CO_2$ fluxes from each of the 15 different TBMs.

Overall, we have difficulty detecting the unique contributions of many environmental driver datasets to variability in the OCO-2 observations. This issue is highlighted by an examination of colinearity in the regression model. In a regression, we cannot estimate different coefficients ($\boldsymbol{\beta}$) for two predictor variables (i.e., columns of $\mathbf{X}$) that are identical or nearly identical; the regression cannot be used to estimate unique coefficients because the predictor variables themselves are not unique. In regression modeling, this phenomenon is known as colinearity. The coefficients ($\boldsymbol{\beta}$) estimated for colinear variables are often unrealistic, and the standard errors or uncertainties in those coefficients ($\mathbf{V}_{\hat{\beta}}$) are often unexpectedly large (e.g., Ramsey and Schafer, 2012). Model selection is one way to reduce or remove colinearity; colinear variables, by definition, do not contribute unique information to a regression and are therefore rarely selected using a model selection approach like the BIC. One common method for detecting colinearity is to estimate the correlation coefficient ($r$) between different columns of $\mathbf{X}$; a value greater than ~0.55 can indicate the presence of colinearity (e.g., Ratner, 2012).

We find substantial colinearity in the regression analysis (Fig. 2). This colinearity likely plays an important role in the model selection results, in addition to errors in the OCO-2 observations and errors in the GEOS-Chem model; it represents and important but potentially overlooked challenge in relating satellite-based $CO_2$ observations to patterns in environmental drivers. The environmental driver datasets are passed through the GEOS-Chem model ($h()$) and interpolated the locations of OCO-2 observations as part of the regression ($h(\mathbf{X})$, Eqs. 1 and 3). In other words, these driver datasets are input into GEOS-Chem in place of a traditional $CO_2$ flux estimate. This step is necessary so that the environmental driver datasets can be directly compared against patterns in the OCO-2 observations. The driver datasets (i.e., columns of $\mathbf{X}$) that we use in the regression are generally unique from one another (i.e., have unique spatial and temporal patterns). However, the differences among many driver datasets disappear once those datasets have been passed through GEOS-Chem. Fig. 2 displays the correlation coefficients

($r$) among environmental driver datasets from MERRA-2 for temperate forests, both before (Fig. 2a) and after (Fig. 2b) those driver datasets have been passed through GEOS-Chem and interpolated to the OCO-2 observations. The correlation coefficients increase substantially from the former to the latter case. This colinearity is independent of errors in the OCO-2 observations and indicates a hard limit on the number of relationships with environmental driver datasets (i.e., coefficients) that we can quantify in the regression. In other words, model selection results are at least partially limited by the limited sensitivity of OCO-2 observations to variations in these environmental driver datasets – either due to atmospheric smoothing and/or due to limitations in the availability of OCO-2 observations in some regions of the globe. This limitation is in addition to the uncertainties due to errors in the OCO-2 observations and the GEOS-Chem model, which also have a critical impact on inferences about $CO_2$ fluxes using OCO-2 observations (e.g., Chevallier et al., 2007, 2014; Miller et al., 2018).

## 3.2 Environmental relationships inferred using observations from OCO-2

We are able to quantify the relationships between OCO-2 observations and several key environmental driver datasets. Figs. 3 and 4 displays the estimated coefficients from the regression analysis using observations from OCO-2. Across extratropical biomes, PAR is the most commonly selected variable. This result reflects the fact that light availability is a key factor that drives $CO_2$ flux variability in mid-to-high latitudes (e.g., Fang and Michalak, 2015; Baldocchi et al., 2017). As expected, the estimated coefficients for PAR are negative, indicating that an increase (or decrease) in PAR in the model is associated with a decrease (or increase) in NEE, indicating an increase (or decrease) in carbon uptake. Note that in this study, negative values for NEE refer to $CO_2$ uptake while positive values refer to net $CO_2$ release to the atmosphere.

By contrast, precipitation and scaled temperature are the most commonly selected environmental driver datasets across tropical biomes. The magnitude of the coefficients for each of these two variables is similar in most biomes, indicating that patterns in both have similarly important associations with patterns in $CO_2$ fluxes. Specifically, a negative coefficient assigned to scaled temperature indicates that an increase in air temperature is associated with increased carbon uptake when air temperatures are cool and reduced carbon uptake when air temperatures are hot; the scaled temperature function has the shape of an upside down parabola, and temperature thus has a different association with $CO_2$ fluxes depending upon whether the air temperature is above or below the optimal temperature for photosynthesis (e.g., Fig. S1). Indeed, high temperatures in the tropics often exceed the optimal temperature for photosynthesis (e.g., Baldocchi et al., 2017), which reduces carbon uptake (e.g., Doughty and Goulden, 2008). Furthermore, negative coefficients for precipitation indicate that an increase in precipitation is associated with an increase in carbon uptake, which is in line with current knowledge that water availability facilitates photosynthesis across seasonal to annual temporal scales, especially in arid or semiarid regions (e.g., Gatti et al., 2014; Jung et al., 2017).

In addition to this regression analysis, we use a GIM to estimate the stochastic component of the fluxes ($\zeta$, Eq. 1) – patterns in the fluxes that are implied by the OCO-2 observations but do not match any existing environmental driver dataset. To this end, Fig. 5 shows the mean contribution of each environmental driver variable and the stochastic component to the GIM across years 2015 – 2018 using MERRA-2 for the environmental driver datasets. The magnitude of the stochastic component in this plot is small relative to the contribution of different environmental variables and relative to the contribution of anthropogenic sources. Furthermore, the stochastic component contains very diffuse spatial patterns, and these very broad patterns do not

imply any clear deficiency in the other components of the GIM. For example, the regression component of the GIM ($\mathbf{X}\hat{\boldsymbol{\beta}}$)

accounts for 89.6% of the variance in the estimated fluxes, and the stochastic component conversely accounts for only 10.4%

of the flux variance. Furthermore, the regression component, when passed through the GEOS-Chem model, matches OCO-2

observations nearly as well as the full posterior flux estimate (Figs. S2 and S13). This result shows that a limited number of

environmental driver datasets can adeptly reproduce broad patterns in $CO_2$ fluxes across continental and global spatial scales

but reinforces the conclusion that current OCO-2 observations are not sufficient to disentangle more complex environmental

relationships.

    In all of the simulations using OCO-2 observations, we estimate a scaling factor ($\boldsymbol{\beta}$) for anthropogenic, biomass burning, and

ocean fluxes of near one, indicating that these source types have a magnitude that is broadly consistent with atmospheric obser-

vations. Specifically, the estimated scaling factor estimated ranges from 0.97 to 1.05, depending upon the year and simulation.

Note, however, that we estimate a single scaling factor for all of these source types combined and are unable to confidently

constrain separate scaling factors for each source, a topic discussed in greater detail in the Supplemental Sect. S2.

    Note that the inferences described here are also broadly consistent with independent, ground-based atmospheric observa-

tions. We specifically model atmospheric $CO_2$ using fluxes estimated from the GIM and compare against regular aircraft

observations, campaign data from the Atmospheric Tomography Mission (ATom; Wofsy et al., 2018), and observations from

Total Carbon Column Observing Network (TCCON; Wunch et al., 2011). In most instances, the model result matches the

observations to within the errors specified in the inverse model (i.e., to within the errors specified in the $\mathbf{R}$ covariance matrix),

and the model-data comparisons do not exhibit any obvious seasonal biases. Furthermore, we also model $XCO_2$ using the

outputs of the regression analysis ($\mathbf{X}\hat{\boldsymbol{\beta}}$), and these outputs also show good agreement with OCO-2 observations (Fig. S13).

The Supplemental Sect. S4, Figs. S2-S13, and Tables S2-S3 describe these comparisons in greater detail.

**3.3   Comparison between inferences from OCO-2 and TBMs**

The environmental relationships (i.e., coefficients) estimated for the TBMs show a substantial range (Figs. 3 and 4); this spread

highlights uncertainties in state-of-the-art TBMs and indicates that there is an opportunity to help inform these relationships

using atmospheric $CO_2$ observations. On one hand, we are only able to infer a limited number of environmental relationships

using current observations from OCO-2, and this fact limits the extent to which we can inform TBM development using

available space-based $CO_2$ observations. On the other hand, we can infer relationships with several key environmental drivers

(e.g., Fig. 3), and TBMs disagree on relationships with even these key drivers. This result thus indicates both the limitations of

this analysis but also its strengths. Specifically, Figs. 3 and 4 graphically displays these results from the regression analysis –

the coefficients estimated using OCO-2 observations compared to those estimated from the TRENDY models. The coefficients

from the OCO-2 analysis are almost always within the range of those estimated using the ensemble of TBMs. With that said,

the coefficients estimated for many of the TBMs are far from the value estimated using OCO-2, implying that observations

from OCO-2 can be used to inform the relationships within numerous individual models. Note that in Figs. 3 and 4, the x-axis

is ordered based upon the environmental driver variables that are selected using OCO-2, and we show the estimated coefficients

for TBMs in which the listed environmental driver variable is also chosen using model selection. Furthermore, the coefficients

shown for the TBMs in Figs. 3 and 4 are calculated using environmental driver datasets from CRUJRA. Fig. S14 displays the results for the TBMs using environmental driver data from MERRA-2, and the results look similar to those using CRUJRA.

We specifically find large differences between the analysis using OCO-2 and the TBMs for relationships with precipitation. The relationships between precipitation are arguably more uncertain within the TBMs than the relationships with other environmental variables (Fig. 3a) and are more uncertain in tropical biomes than temperate ones. This statement is particularly apparent when we examine the coefficient of variation for each relationship (Fig. 3b). The coefficient of variation is a measure of the uncertainty relative to the magnitude of the mean, and Fig. 3b shows the standard deviation in the coefficients from the 15 TBMs divided by the mean coefficient from the ensemble of TBMs. In addition, the TBMs are evenly split on whether the relationship with precipitation is positive or negative across tropical biomes, and our analysis using OCO-2 observations agrees with models that estimate a negative relationship (i.e., precipitation is associated with greater $CO_2$ uptake). There is substantial disagreement on the magnitude of this relationship, even among models that yield a negative relationship; the estimate using OCO-2 observations falls in the mid-range of these TBMs for both tropical biomes.

More broadly, the TBMs simulate very different water cycling through each ecosystem, in spite of the fact that each model uses the same precipitation inputs from CRUJRA. These broader differences in water cycling within the TBMs may help explain the large uncertainties in the relationships between $CO_2$ and precipitation, and highlights an important source of uncertainty within these models. Specifically, we find that estimated evapotranspiration (ET) across the TBMs differs by almost a factor of three among models in some seasons and biomes, and annual ET ranges from 375 mm to 700 mm over North Hemispheric tropical grasslands (Fig. 6a), and from 530 mm to 1010 mm over North Hemispheric tropical forests (Fig. 6b). These large differences in ET estimates reinforce the very different responses of tropical ecosystems in these models (both tropical forests and tropical grasslands) to precipitation inputs.

Indeed, existing studies have indicated large uncertainties in the responses of tropical forests to water availability (e.g., Restrepo-Coupe et al., 2016) and have offered several possible explanations. Soil depths and rooting distribution are particularly challenging to model in tropical ecosystems, yielding uncertainties in the relationship between water availability and $CO_2$ fluxes (e.g., Baker et al., 2008; Poulter et al., 2009). For example, Poulter et al. (2009) argued that current TBMs tend to underestimate soil depths in tropical forests, which are critical to guarantee soil water access and to accurately simulate dry-season photosynthesis in TBMs. The treatment of irrigation and other land management practices also differs among models and creates further uncertainty (e.g., Le Quéré et al., 2018; Pan et al., 2020). To complicate matters, the role of precipitation in carbon dynamics can vary depending on broader environmental conditions and the time scales considered (e.g., Baldocchi et al., 2017). For example, excess precipitation is associated with limited light availability in regions like the humid tropics and can raise the water table to a level that inhibits respiration. With that said, short-term rain events have been shown to boost respiration (e.g., Baldocchi, 2008).

Like precipitation, relationships with PAR are also highly uncertain in the simulations using TBMs. Most models yield relationships with the same sign, but those relationships vary widely in magnitude. By contrast, results using OCO-2 observations are very similar to the ensemble mean of the TBMs. This result is particularly interesting given that the individual TBMs do not show consensus with one another. The differences among the TBMs likely stem from the fact that these TBMs exhibit widely

varying seasonal cycles and peak growing season uptake across extratropical biomes. For example, in temperate forests (e.g., Fig. S17), the maximum monthly carbon uptake differs by a factor of eight among the TBMs, and a handful of TBMs estimate a very different seasonal cycle than the bulk of the TBMs with maximum uptake during the middle of the growing season.

In contrast to the discussion of precipitation and PAR, existing TBMs yield much better agreement on the relationships between $CO_2$ and scaled temperature (Fig. 3). In tropical biomes, nearly all TBMs agree on the sign of the relationship, and the estimates using OCO-2 observations are within the range of those estimated using TBMs. Interestingly, the uncertainty bounds on the coefficients estimate using OCO-2 are not that much smaller than the range of coefficients from the ensemble of TBMs, both for tropical grasslands and especially for tropical forests. This result points to relatively good consensus in modeled relationships with temperature for tropical grasslands and forests – both using TBMs and OCO-2 observations. However, it also indicates that atmospheric observations from OCO-2 potentially have less opportunity to inform these relationships than for precipitation or PAR where TBMs do not show consensus.

The comparisons described above are largely from biomes centered in the tropics and mid-latitudes and include few comparisons for high latitude biomes (e.g., the boreal forest or tundra biomes). For example, we do not select any environmental driver variables for the tundra biome using OCO-2 and only select PAR in boreal forests. OCO-2 observations are sparse across high latitudes both due to the lack of sunlight in winter and due to frequent cloud cover in many high latitude regions. We also only select PAR in boreal forests in simulations using two of the 15 TBMs. This result also reflects the limited availability of OCO-2 observations over high latitude regions; for the analysis here, we create synthetic OCO-2 observations using each TBM and apply model selection to each of these synthetic OCO-2 datasets. Hence, the sparsity of OCO-2 observations not only affects the model selection results using real OCO-2 observations but also affects the analysis shown in Figs. 3 and 4 using the TBMs. The fact that PAR is selected for so few TBMs is not a reflection on the important role of PAR across the boreal forest in many TBMs.

Note that the analysis described above is based upon the mean relationships that we infer for years 2015–2018. We also explored how these relationships in the models vary during El Niño (2015–2016) and non-El Niño years (2017–2018) (Fig. 4). The relationships that we estimate do not fundamentally change between El Niño and non-El Niño years and neither does the spread among the models. This result indicates two conclusions: (1) there is not a fundamental shift in these relationships between El Niño versus non-El Niño years, suggesting that it is not the change in environmental relationships but the change in environmental variables themselves that correlate with the change in flux estimates; and (2) the uncertainties in the relationships, as estimated by the TBMs, are not higher in El Niño versus non-El Niño years. With that said, the magnitude of the estimated coefficient does change in some models between El Niño and non-El Niño years; the changes in the coefficients are generally less than 50% in most models, and the models do not show a consistent direction of change between El Niño and non-El Niño years.

# 4 Conclusions

In this study, we use four years of observations from OCO-2 and a top-down statistical framework to evaluate the relationships between patterns in atmospheric $CO_2$ observations and patterns in environmental driver datasets that are commonly used in modeling the global carbon cycle. We are able to quantify a limited number of these environmental relationships using observations from OCO-2. In spite of these limitations, we are still able to identify relationships with a small number of salient environmental drivers datasets, and state-of-the-art TBMs do not show consensus on some of these key relationships, indicating an opportunity to inform these relationships using atmospheric $CO_2$ observations.

We subsequently compare inferences using OCO-2 against inferences from 15 state-of-the-art TBMs that have model outputs available for the same set of years. For the broad regions and timespan explored in this study, we find negative relationships between patterns in OCO-2 observations and patterns in precipitation; this result agrees with half of the TBMs, which do not show consensus on relationships with precipitation. By contrast, TBMs exhibit much greater skill in describing relationships with scaled temperature, as implied by the relatively good agreement among TBMs. In fact, the uncertainties in the temperature relationship across tropical biomes, as estimated using OCO-2 observations, is nearly as large as the range of estimates using TBMs.

More broadly, state-of-the-art TBMs disagree on the contribution of individual biomes to the global carbon balance, a result highlighted in several studies (e.g., Poulter et al., 2014; Sitch et al., 2015; Ahlström et al., 2015; Piao et al., 2020). In order to reduce these uncertainties, scientists will likely need to reconcile differences in the environmental processes that drive these $CO_2$ flux estimates. Existing studies have used in situ atmospheric observations to help quantify and evaluate these relationships across the extratropics (e.g., Fang and Michalak, 2015; Hu et al., 2019). However, this task is much more challenging across regions of the globe with sparse in situ observations, including most of the tropics. In spite of the limitations described in this study, the advent of satellite-based $CO_2$ observations like those from OCO-2 provide a new opportunity to constrain these environmental relationships and thereby provide unique atmospheric constraints on the global carbon cycle.

*Data availability.* The version 9 of 10-s average OCO-2 retrievals are available at ftp://ftp.cira.colostate.edu/ftp/BAKER/; data information of the OCO-2 MIP is available at https://www.esrl.noaa.gov/gmd/ccgg/OCO2/; data information of the ObsPack data product is available at http://www.esrl.noaa.gov/gmd/ccgg/obspack/.

*Author contributions.* Z.C. and S.M.M. designed the study, analyzed the data, and wrote the manuscript. S.S., P.F., V.B., D.S.G., V.H., A.J., E.J., E.K., S.L., D.L.L., P.C.M., J.R.M., J.E.M.S.N., B.P., H.T., A.J.W, and S.Z. provided TRENDY model flux estimates. All authors reviewed and edited the paper.

*Competing interests.* The authors declare they have no competing interests.

*Acknowledgements.* Financial support for this research has been provided by NASA ROSES grant no. 80NSSC18K0976. We thank Kim
Mueller and Anna Michalak for their feedback on the research; David Baker for his help with the $XCO_2$ data; Colm Sweeny and Kathryn McKain for their help with aircraft datasets from the NOAA/ESRL Global Greenhouse Gas Reference Network; John Miller, Luciana Gatti, Wouter Peters and Manuel Gloor for their help with the aircraft data from the INPE ObsPack data product; Steven Wofsy, Kathryn McKain, Colm Sweeny, and Róisín Commane for their help with ATom aircraft datasets; and Debra Wunch for her help with the TCCON datasets. Daven Henze's work is supported by NOAA grant no. NA16OAR4310113. Daniel Goll's work is supported by the ANR CLAND
Convergence Institute. The data analysis and inverse modeling were performed on the NASA Pleiades Supercomputer.

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

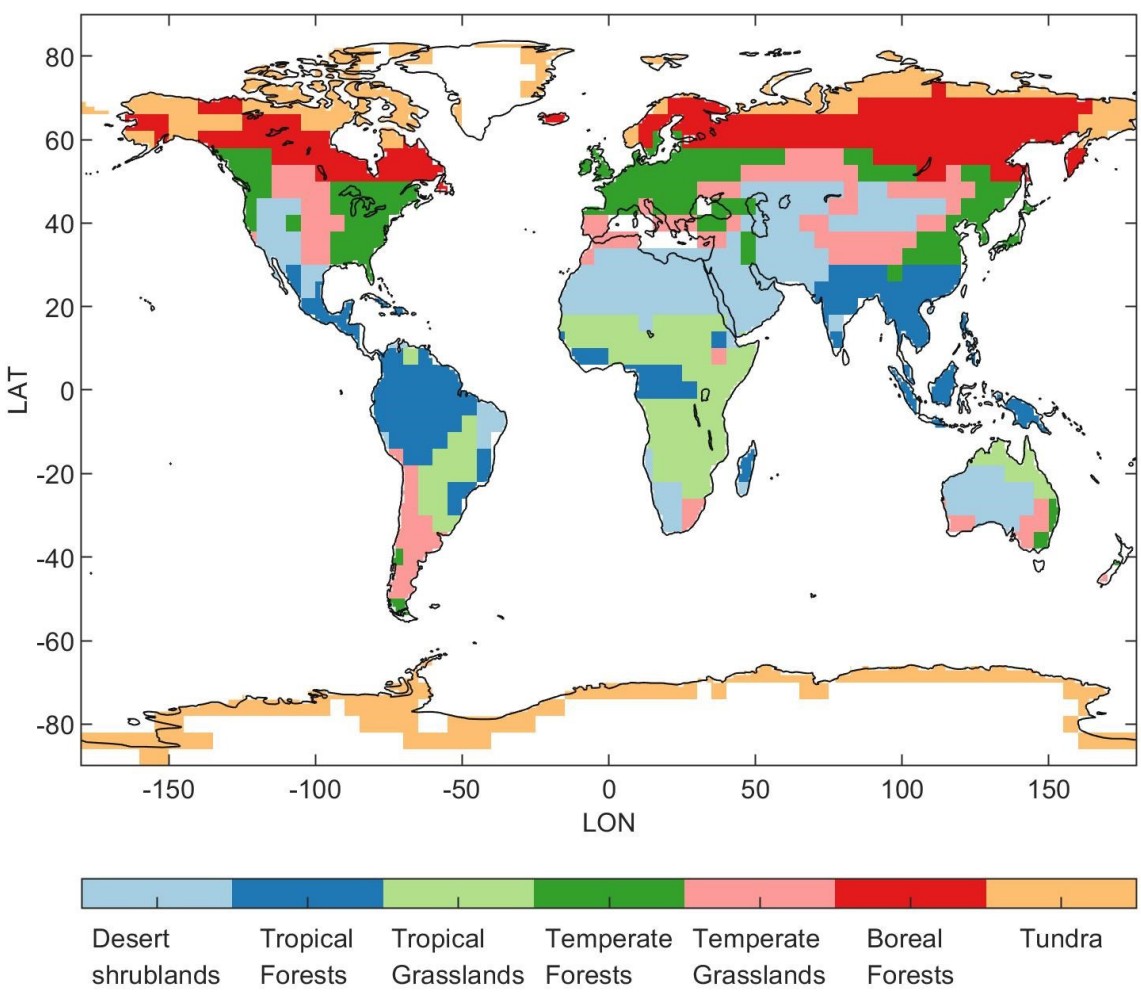

**Figure 1.** The seven biome-based regions aggregated from a world biome map in Olson et al. (2001).

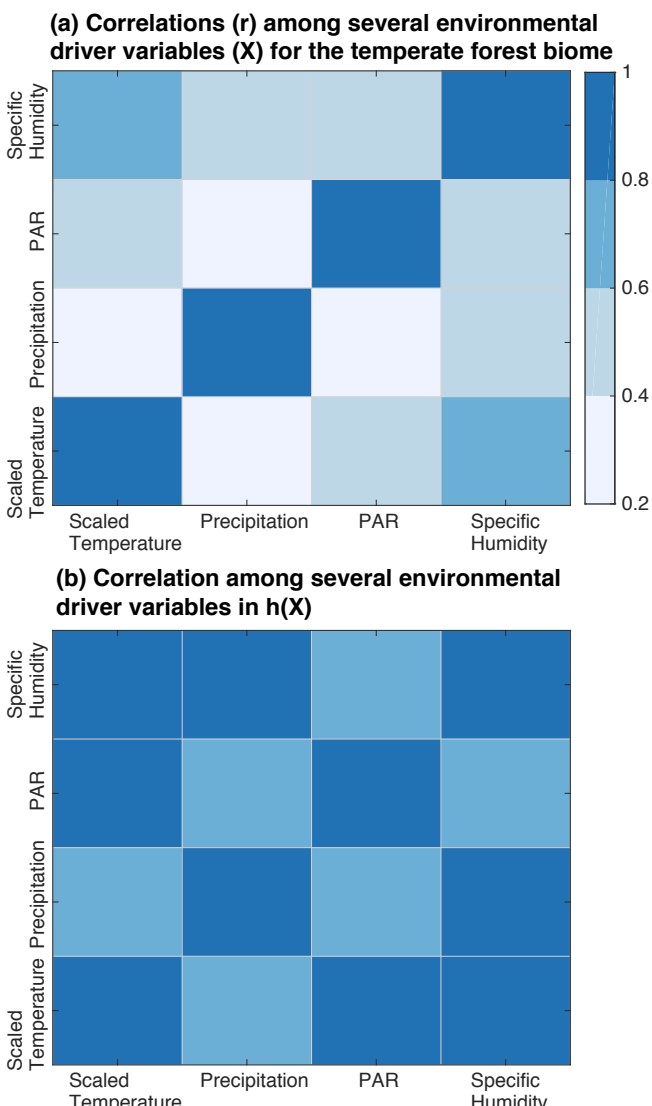

**Figure 2.** Correlation coefficients (r) between environmental drivers over temperate forests biome in year 2017 in $\mathbf{X}$ (panel a) and $h(\mathbf{X})$ (panel b). We find that the correlation between environmental drivers ($\mathbf{X}$) are generally low (a), e.g., PAR and scaled temperature, precipitation and specific humidity; however, when these environmental drivers are passed through the transport model $h()$ and interpolated to the locations of OCO-2 observations, the correlation between these drivers become much stronger (b), indicating high collinearity.

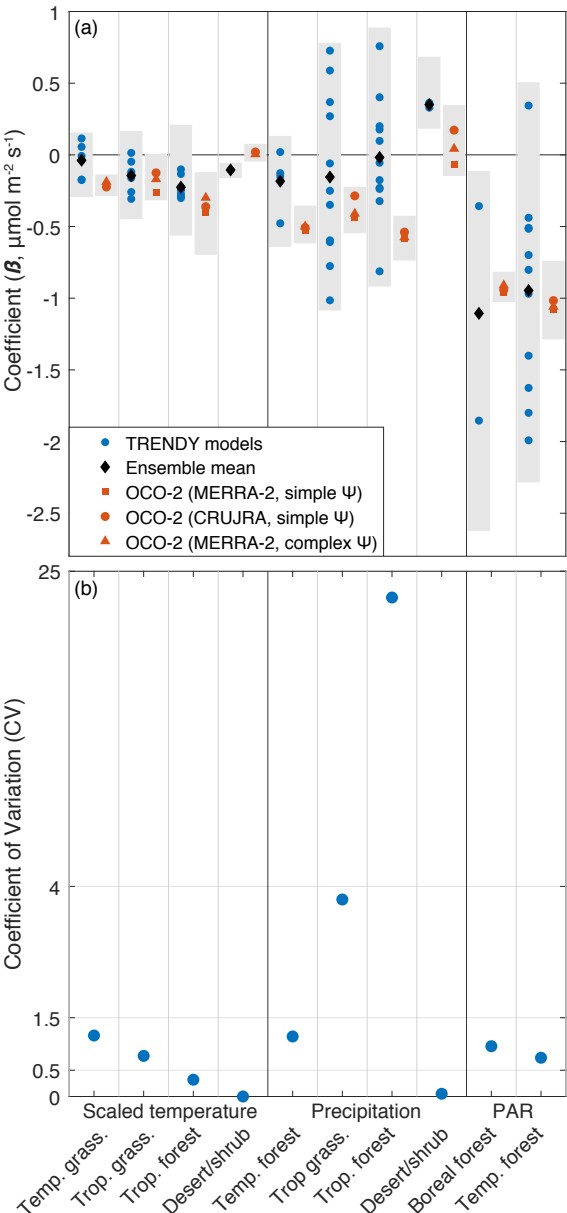

**Figure 3.** Estimated coefficients ($\beta$) from the TRENDY models (blue), from the ensemble mean of the TRENDY models (black), and from the analysis using OCO-2 (red). Each blue or red dot indicates the mean value across all four years of the study period. Gray bars indicate the full range of uncertainties in the coefficients. To construct these gray bars, we calculate the uncertainties in the coefficients estimated for each individual TBM (or for the real OCO-2 data) using Eq. 4. They gray bars encapsulate all of the uncertainty bounds from all of these individual model calculations. Furthermore, the analysis of OCO-2 includes simulations using MERRA-2 meteorology with a simple formulation of $\Psi$ (red square), using CRUJRA meteorology and a simple formulation of $\Psi$ (red dot), and using MERRA-2 and a complex formulation of $\Psi$ (the same used in the GIM, red triangle). The coefficients from the analysis using OCO-2 (red) are broadly within the range of the estimates in TBMs (blue). We further calculate the coefficient of variation (CV) of coefficients for each environmental drivers within the TBMs (b), and we find that the largest CV are from the coefficients for precipitation.

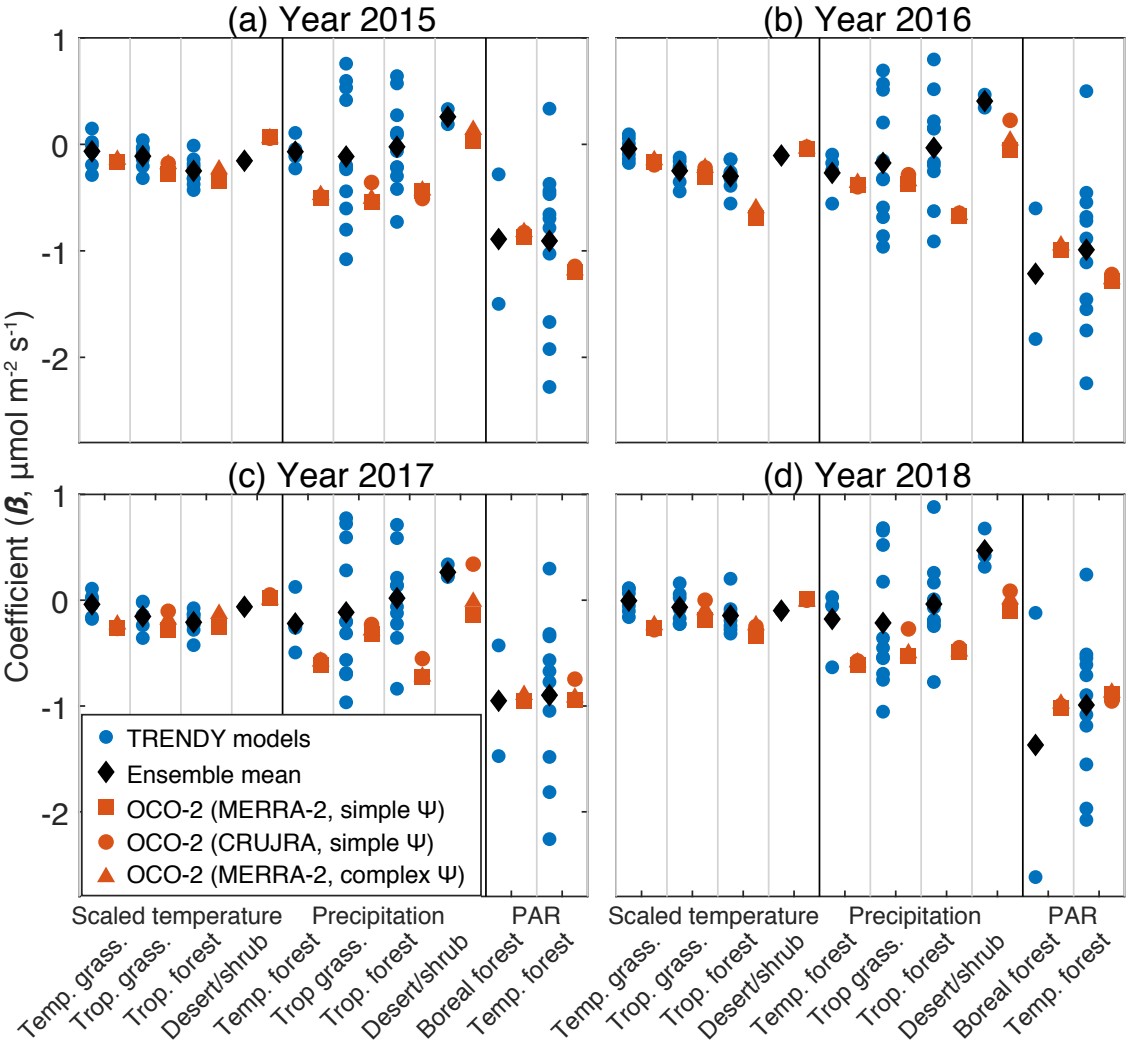

**Figure 4.** This figure is similar to Fig. 3 but shows results for individual years. There are no noticeable shifts in the coefficient estimates between El Niño (2015 – 2016; panels a-b) and non-El Niño years (2017 – 2018; panels c-d) from the analysis using OCO-2 (red). Some individual TBMs show differences of up to 50% in the estimated coefficient among years, though many individual TBMs do not.

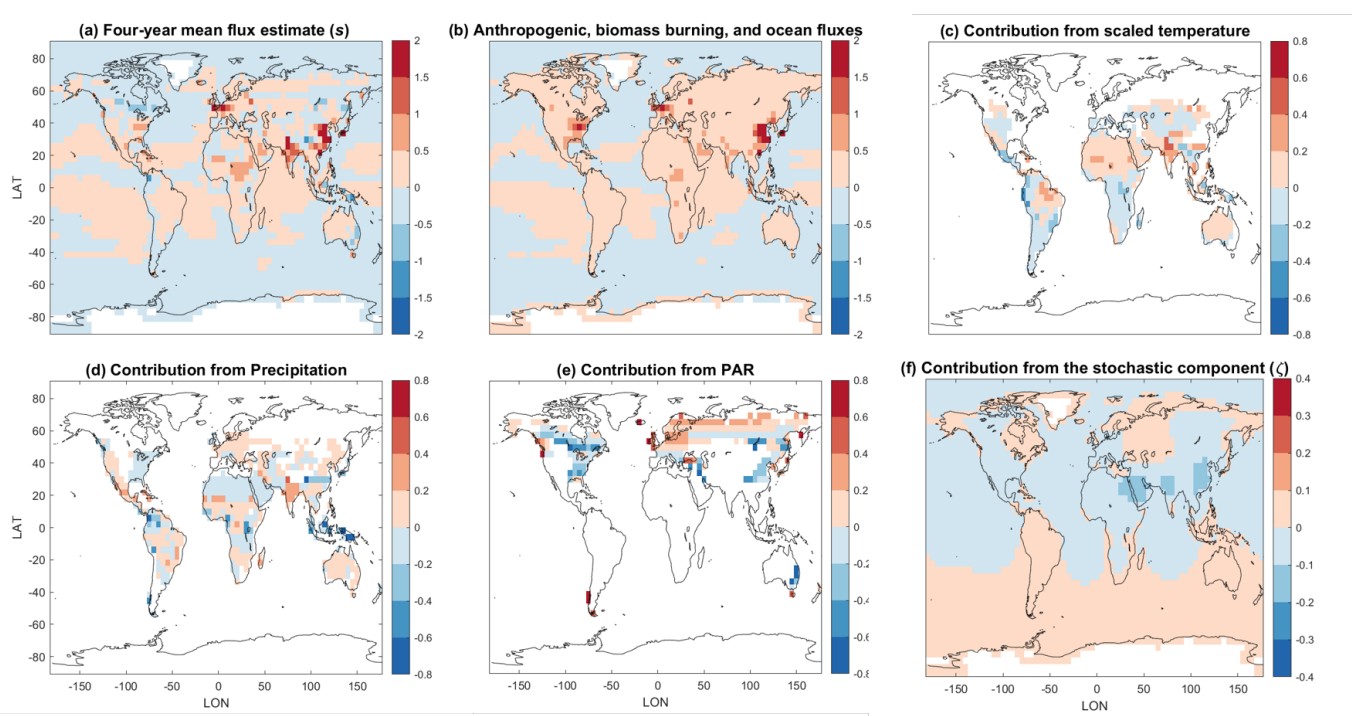

**Figure 5.** The contribution of different environmental driver datasets to the flux estimate from the GIM. Panel (a) displays the four-year mean flux estimate (including both the regression and stochastic components of the flux estimate; units of $\mu$mol m$^{-2}$ s$^{-1}$) and panel (b) the contribution from anthropogenic, biomass burning, and ocean fluxes. Contributions from different environmental drivers, including scaled temperature (c), precipitation (d), and PAR (e), describe most of spatiotemporal variability in terrestrial biospheric CO$_2$ fluxes, whereas the stochastic components ($\zeta$), panel f) only account for a small portion of flux variability. Note that the inverse modeling results shown in this figure use environmental driver data from MERRA-2. Also note the color bars used in panels (a-b), panels (c-e), and panel (f) are different. White colors in panels (c-e) indicates that not all environmental drivers are selected in all biomes.

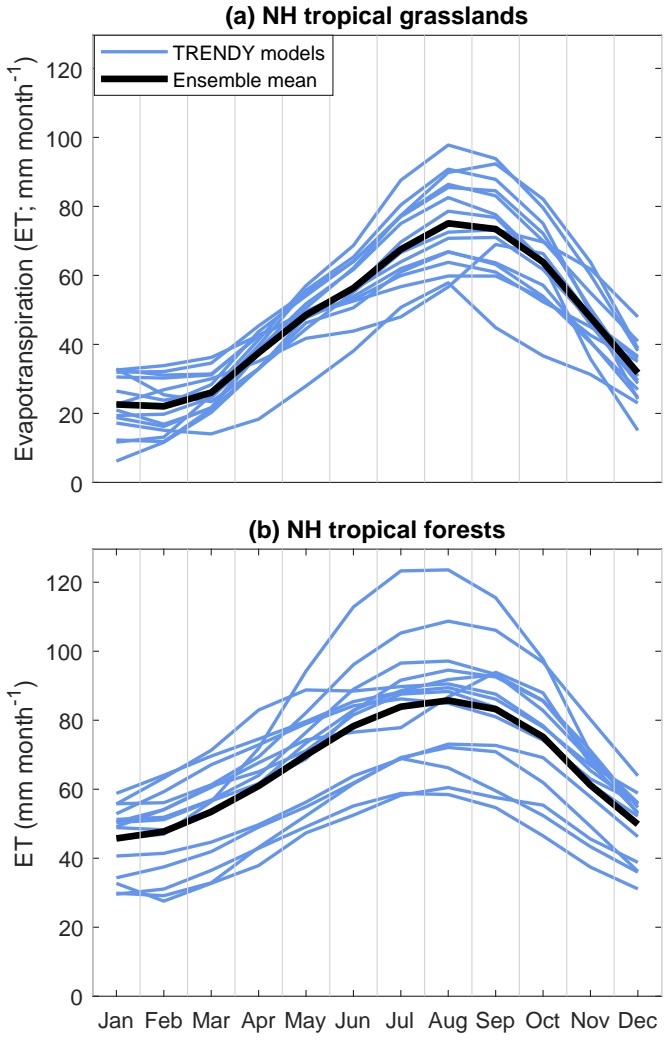

**Figure 6.** Four-year averaged evapotranspiration (ET) estimates from a suite of 15 TBMs (blue) and from the ensemble mean (black), for North Hemispheric tropical grasslands (a) and for North Hemispheric tropical forests (b). Annual ET show large differences in magnitude across the TBMs for both tropical biomes.