# Peer review of "Linking global terrestrial CO2 fluxes and environmental drivers: inferences from the Orbiting Carbon Observatory-2 and terrestrial biospheric models"

_Atmospheric Chemistry and Physics, 2020_

## Referee Comment (RC1) · Anonymous Referee #1 · 26 May 2020

The authors have developed a geostatistical inverse method to interpret satellite observations of carbon dioxide (CO2) collected by the NASA Orbiting Carbon Observatory collected during 2016. As far as this reviewer can see the study is scientifically sound but describes only an incremental improvement to the method and does not lead to any new scientific insight. Unfortunately, the authors' choice of OCO-2 data raises more questions than it answers.

Major comments

The environmental drivers for ecosystems located at mid/high and tropical ecosystems are unsurprising. Perhaps that's the point. I wasn't sure. PAR is by definition photo-

synthetic active radiation so its ability to describe large-scale CO2 fluxes isn't anything new, particularly over one year that is dominated by the seasonal cycle. Any insights from using the diffuse and direct components of PAR? Similarly, temperature and precipitation roles in the tropics are nothing new. However, I am surprised that precipitation is such a useful driver over the tropics where complex basin-scale hydrologic controls are at play. In other words, where it rains is not necessary where the water ends up.

The authors have gone some way to 'fess up that the geostatistical inverse method uses prior information for which I commend them. It might not be defined in the same way as the classical Bayesian approach but nonetheless it uses prior information. Otherwise, inferring fluxes for 10ˆ6 grid boxes using 10ˆ5 measurements is an ill-posed problem. The method uses environment driver data with uncertainties that are difficult to quantify (see comment below about estimated posterior uncertainties).

It would be useful to reiterate to the reader the benefit of the geostatistical inverse method over more traditional methods. Certainly, it provides an alternative perspective but I have seen no evidence to suggest it is better or worse.

Line 216: This reader is surprised that OCO-2 data are not sensitive to biomass burning emissions, particularly during the El Nino period. The manuscript would benefit from having more explanation on this point.

Why are correlations higher when environmental drivers are passed through the atmospheric model. Figure 3 doesn't cut it - the color scale is almost binary as currently defined. Using the square of the correlation might be a better way to illustrate these calculations.

Line 263: widespread and prolonged drought conditions, together with large-scale land-use change, is a more accurate description of what's going on over these regions.

Paragraph 298: comparison of the reported work and other groups is weak. Not many people have used v9 of OCO-2 data so I think it would be useful for the readership

to provide a more detailed assessment of results compared with past estimates using v7 data. The comparison between the model and independent measurements is minimal (in supplementary information). The uncertainties associated with the posterior estimates are unrealistically small. The classical Bayesian inversion as typically employed underestimates posterior uncertainties so certainly the uncertainties estimates reported with the geostatistical method are grossly underestimated. This reviewer is left wondering why this might be so and how a possible explanatory imbalance between prior and observation uncertainties would influence model selection and the analysis that follows.

Sure, the tropical flux estimates are important to discuss. However, are the reviewers are in a position to dismiss the results over tropical North Africa without further explanation. Why did they find themselves in terms of environmental drivers? Surely, their results over tropical Africa aren't exclusively determined by measurements collected over tropical Africa? Do they find that seasonal differences in measurement over tropical Africa lead to a bias in the flux? Answers to these questions would represent a useful contribution to the field.

There is almost nothing in the manuscript about the large differences between other geographical areas where we would expect much better agreement, e.g temperature North America, Europe, Eurasian temperate. Without a more comprehensive evaluation of the fluxes it is difficult to know whether the method is at fault or the data they have used. This manuscript would benefit greatly from a better evaluation of the posterior fluxes.

Minor comments

Line 59: it would be fairer that Chevallier 2018 argues not suggests.

For context, it would be useful for the reader to understand that 2016 was an El Nino year.

[Figure]

Line 91: how did the authors decide that four months was a sufficient spin-up period?

---

## Short Comment (SC1) · 18 Jul 2020

In this work, the authors use a geostatistical inverse modelling approach to infer surface fluxes from observations of column CO2 by the Orbiting Carbon Observatory 2 (OCO-2). Using these estimates, the authors make claims about the environmental drivers of the spatiotemporal variability of surface fluxes. However, their evaluation against independent data (sometimes coarsely defined as "validation") is not sufficient to support these claims.

Inferring surface carbon fluxes from observations of atmospheric CO2 is an inherently ill-defined problem. Its solution, in any form, requires a number of assumptions that

are often poorly constrained by existing scientific knowledge. The authors do a commendable job of explaining that despite erroneous claims in the existing literature to the contrary, geostatistical inverse models do in fact use prior information, just in a different form than more common approaches. What the authors fail to do is support that their surface flux estimates are fit for the scientific purpose at hand. Typically, this is accomplished through comparisons to other independent data products. While pedantic, it seems more and more necessary that we remind ourselves that inferred surface fluxes fall into the prediction step of the Scientific Method. Between that and the analysis step, is the all important testing step. The testing step cannot be shortcut – it is the only thing separating science from plausible guesswork.

In order to make claims about the spatiotemporal variability of surface fluxes, the authors must first evaluate the fidelity of their surface fluxes' spatiotemporal variability. While this reviewer admits that there is no ideal method of evaluating global surface fluxes of CO2 on horizontal scales greater than a few tens of kilometers, a greater effort must be made to demonstrate the product is appropriate for the analysis in the text. In particular, the only evaluation of their surface fluxes is that of long-term time mean regional budgets (Figure 6) and simulated CO2 at just a handful of aircraft profiling sites (Figures S5 and S6). If one is to make claims about seasonal cycles, for example, then the seasonal cycle of the inferred fluxes must be evaluated as well. Given the assumptions necessary to make these inferences, it is entirely possible that their long-term time mean budgets are reasonable and their seasonal cycles are not. This is especially important given the documented impact (Basu et al., 2013, ACP; Crowell et al., 2019) that very small seasonal and regional biases from satellite retrievals can have on inferred fluxes. Unless the authors are able to demonstrate the skill of their product in reproducing variations over the same spatiotemporal scales as the scientific analysis, this review does not see how their claims can be supported.

———————————————

---

## Referee Comment (RC2) · Julia Marshall (Referee) · 27 Jul 2020

At first glance it seems that the results of this study make sense, and are consistent with our general understanding of what drives carbon fluxes, with uptake at higher latitudes being mostly radiation-limited while in the tropics there are more complex temperature-precipitation interactions. So far, so good. The paper is well written and clearly structured, making it easy to read.

To the careful reader it soon becomes clear that something is going wrong, however, and the limited "validation" and comparison to other results from the literature are insufficient to explain these problems away. While the geostatistical approach is com-

mendable in that it allows more flexibility in the structure of the prior fluxes, such that perhaps unexpected signals may emerge, it also seems to allow for rather unphysical results, as in this case. Given the fact that the ocean fluxes (a net sink of more than 2 PgC/year) were rejected by the Bayesian Information Criterion (BIC) while the net land fluxes are more or less consistent with other studies, it seems impossible that the global atmospheric growth rate can be matched. It just does not add up.

This should be obvious when performing validation, but the very little testing of the posterior fluxes, limited to a handful of aircraft measurements far from coasts on a scatter plot averaged (monthly?) by height, hidden in the supplement, makes it hard to tell. The paper states that aircraft profiles near coasts were not used because the coarse model resolution made it hard to represent these data well, but I wonder if the complete absence of ocean fluxes may have also played a role here?

Since none of the in-situ sites were used for constraining the fluxes (which seems an odd choice, even if only for comparison's sake), it would be instructive to plot the concentrations resulting from the posterior fluxes at a few sites to see if the curves drift apart over the year as a result of the missing sink. While this might not look too bad in a simulation of only one year, this would soon result in wildly divergent curves. But perhaps over a longer simulation the BIC would then choose to select the ocean fluxes. Still, the decision to blindly allow the model to return what we know is incorrect makes it hard to trust the interpretation of the results. Perhaps Takahashi was not the best ocean prior in this case, especially for an El Niño year, and this played a role: this could be an area for more analysis.

The comparison to other model output was largely limited to the OCO-2 model inter-comparison study of Crowell et al. (2019), without following the considerable effort they put into validation or consideration of in-situ measurements. Looking at TCCON sites is an obvious choice, as is the extension to additional aircraft measurements, such as AToM, which are available for at least a couple months of 2016. But comparing your (unclosed) budget to the land biosphere budget of other (mass-conserving) studies is

intrinsically misleading. (I am not as surprised that BIC did not pick out the GFED emissions, as these are a few orders of magnitude smaller and are easily swallowed up in the biosphere signal.)

As the carbon budget presented in this study does not seem to add up in a basic back-of-the-envelope way, and the validation presented was not sufficient to identify that, I cannot recommend its publication without substantial revisions.

Other major comments:

L10 & L204-205: While the difference in wording is subtle, I think the abstract over-states what the meteorological variables explain. Do they really describe 90% of the variability in the fluxes (as seen through OCO-2 observations)? This sort of implies that OCO-2 can "see" fluxes, which isn't true of course. The latter explanation that the deterministic model accounts for XX% of the variance in the estimated fluxes seems more accurate. As you're only treating fluxes on a daily time scale, you're definitely not describing 90% of the variability in the fluxes themselves.

Figure 3 and discussion around L235: This is actually quite interesting! I would be interested in seeing some more analysis of this point. It was also not entirely clear to me what was correlated (and how) in Figure 3. The meteorological variables have been "passed through an atmospheric [transport] model": were they then sampled as column-averaged variables, as OCO-2 views the atmosphere? Were the same averaging kernels applied? It also says that this is the correlation "within different global biomes". Were these columns averaged across space then, and the correlation taken in time? Or is this a spatial correlation coefficient between the column-averaged maps for a given time? I feel like there is an intriguing result here, but I don't fully understand what you've done.

L238 & 239: How can you be sure that this collinearity is playing a bigger role than retrieval or model errors? Would the latter two effects not also limit the model selection?

L244 & L260: These statements seem to contradict each other. The first says that the negative beta values for PAR mean that an increase in PAR leads to a decrease in NEE and an increase in uptake. The latter says that the negative beta value for scaled temperature means that an increase in temperature leads to reduced uptake. How can these both be true? This is fundamental to the conclusions drawn.

L257-258: While cloudiness is correlated with clouds and rainfall, it's also correlated with the presence or absence of satellite measurements. What impact might this have on your results?

L302: I'm actually surprised Australia matches as well as it does, as you've had to fold the Southern Ocean sink into the Southern Hemisphere land fluxes somehow.

Minor comments:

In several places "in year 2016" should be replaced with "in the year 2016".

L42: refer -> referred

L46: levels -> level

L48: "At" should not be capitalized.

L55: region -> regional

L59 (and elsewhere): the ñ is in italics throughout.

L62: modeling -> model

L73: Here you mention that you are optimizing daily fluxes. Does this mean that the diurnal cycle is completely ignored?

L95: average -> averaged

L114 & 176: space before italic "p"

---

## Referee Comment (RC3) · Abhishek Chatterjee (Referee) · 3 Aug 2020

Review of "***Linking global terrestrial CO$_2$ fluxes and environmental drivers using OCO-2 and a geostatistical inverse model***" by Zichong Chen *et al*.

This study presents the linkages between flux estimates derived from OCO-2 retrievals and environmental drivers across globally 7 biome-based regions. Using a geostatistical inverse modeling (GIM) approach, the authors demonstrate that they are able to identify connections between carbon flux and three key environmental drivers, namely air temperature, daily precipitation and PAR – the combination of these three variables explaining more than 89.6% of the variability in CO$_2$ fluxes. However, the study is conducted for one year only (circa 2016). The authors claim that this is an initial case study, thus implying that a more comprehensive study will follow later. This begs the question – is this study intended to demonstrate that the GIM approach has been successfully adapted to remote-sensing observations (i.e., a technical study) or is it intended to capture the connections between CO$_2$ fluxes and environmental drivers (i.e., a scientific study)? Kindly see Major Comment #1.

I believe the authors ideally wanted it to address a bit of both but unfortunately, in trying to address both, the authors end up addressing neither. I highly recommend that the authors take a step back and decide whether to focus on the inversion methodology and application to OCO-2 retrievals **OR** highlight the scientific questions related to regional and seasonal environmental drivers, and then resubmit. In general, the manuscript is well-written and concise, but it falls short of a clear formulation in terms of scientific scope, depth and novelty.

Several other questions persist. These revolve around limited validation of the posterior flux estimates or posterior CO$_2$ concentrations (see Major Comment #4). The choice of the model-data-mismatch variance (**R**) is inconsistent with real OCO-2 retrievals and needs justification in the main text (rather than bypassing it and relegating it to the Supplementary Section). **R**, along with the a priori flux covariance matrix **Q**, balances the relative weight of the atmospheric data and the trend in estimating the fluxes. An inverse modeling study cannot gloss over these details (see Major Comment #6).

Along with my comments below, I have suggested a few basic analyses and additional experiments, that will improve this study and make it scientifically robust and appealing to the larger carbon cycle science community. I sincerely hope that the authors consider these suggestions.

**Major Comments:**

1) *Scope of the study* – as mentioned earlier, the authors need to lay out a clear scope early on and remain consistent throughout. If the authors are interested in examining the relationship between carbon flux and environmental drivers, a one-year study is not justifiable. The authors need to examine the relationship over a number of years, make sure they are capturing the inter-annual variability in their flux estimates and then assess the relationship between drivers and fluxes. In addition, it is worth noting that the selected year is an El Niño year. On Page 3, Lines 86 – 88, the authors justify this decision by pointing out that the OCO-2 observations had 7-week gap in 2015- and 1.5-month gap in 2017. Remote sensing datasets, or rather any real observations, will always have data gaps! Simply discarding entire years' worth of data for a 5-7-week gap is not a reasonable justification. On the other hand, if the authors want to highlight the development of a new inversion framework/methodology, then it may be out of scope for ACP, and may be better suited to a journal like GMD, where a

lot of the mathematical nuances can be captured. Right now, a lot of the important mathematical details have been relegated to the supplemental material, including important discussions about the error covariance parameters and how they are derived. These details need to be included in the main text.

2) *Scientific novelty* – The authors report that a combination of PAR, daily temperature and daily precipitation are the most adept at capturing the variability in the fluxes (PAR for mid-to-high latitudes and a combination of daily temperature and precipitation for the tropical biomes). Neither of these findings are unique. The authors have correctly referred to a host of studies using GIM (e.g., Gourdji et al. 2008, Fang and Michalak, 2015, among others) or studies using OCO-2 data that have examined the response of the land carbon cycle during the 2015-2016 El Niño (e.g., Liu et al., 2017, Crowell et al., 2019). The BIC did its job and picked up the variables it was supposed to; hence, it is slightly unclear how this study adds new insights into our knowledge about carbon cycle science. In fact, by the authors own admission in Sections 3.1.1 and 3.1.2, almost all their findings are exactly the same as reported in previous studies. These two sections almost read like a literature review rather than a results section with new and exciting science results.

3) *Selection of auxiliary variables and how they are being reported* – what may add a new dimension, relative to already published studies, is reporting a table with all the 12 selected environmental drivers and including the estimated drift coefficients, coefficient of variation, annual average contribution to flux and the correlation coefficient between the selected auxiliary variables in the model of the trend. Actually, the annually averaged global contribution to flux can be reported in typical carbon flux units (like GtC/yr or PgC/yr). That would be novel information, especially if it were to be compared against estimates based on in situ data. Finally, just out of curiosity, why didn't the authors select *fPAR* instead of *PAR*? Also, the authors argument for not including LAI or SIF because they are "remote sensing indices" (Page 5, Lines 144-146) is surprising. Almost all of the auxiliary variables listed on Lines 138-141 are derived from remote-sensing measurements. What if the authors were to include LAI? How would that change their selected model of the trend?

4) *More rigorous evaluation of posterior flux estimates and more importantly, posterior concentrations, against independent measurements* – The biggest surprise of this study is that there are extremely limited evaluations presented against independent measurements (only 7 aircraft sites!). Given the large number of available independent datasets (*in situ* such as surface flask sites, towers and aircraft, TCCON), the absence of a detailed evaluation is striking. Especially, from a seasoned inverse modeling team. Since the authors claim that they are estimating daily global $CO_2$ fluxes at the GEOS-Chem grid scale (Page 3, Lines 72-73), there should be no reason for not evaluating against observations from dedicated aircraft campaigns such as ATom or ACT-America. In addition, it is also not clear why in Section S7, the authors allude to the results from Crowell et al. 2019. The authors have to back up their own biases and RMSD and explain those numbers and their significance, rather than pointing the reader to Crowell et al. 2019 for justification.

5) *Comparison of findings against those derived from in situ data* – The value of this study will be significantly enhanced, if the authors do the same analyses utilizing in situ data (such as

NOAA obspack). Are the conclusions, especially in terms of the three significant drivers and their contribution to the carbon flux, consistent? It has been 12+ years since the Gourdji et al. 2008 study attempted such an analysis – given the increase in the number of surface flask sites and improvements in atmospheric transport model, availability of auxiliary datasets, it will be worth revisiting this and comparing against the information reported here from OCO-2 datasets.

6) *Error covariance parameters* – Can the authors explain why they switched to a spherical covariance model instead of sticking with a simpler exponential covariance model? The authors argue that the shorter correlation length is due to higher density of observations relative to previous studies. Part of that is true. But I believe that the shorter correlation length in the residuals is more reflective of the model of the trend that has been fitted to large biome scales. The model of the trend is too complex for the biome scale; for the grid scale studies that the authors allude to, it made sense. Additionally, the authors persist with a model-data mismatch variance of 1.19 ppm$^2$ based on a previous pseudo-data study. Why? I highly encourage the authors to use the reported $X_{CO2}$ uncertainty for the OCO-2 soundings and then add reasonable representation of transport and representation errors to get 'real' MDM variances. This shouldn't be a huge task given the involvement of core GEOS-Chem developers in this study. It wouldn't be surprising if more reasonable **R** values lead to an increase in *a posteriori* uncertainties for their flux estimates (Page 11, Lines 324-325).

--
Abhishek Chatterjee
Global Modeling and Assimilation Office (GMAO)
Goddard Earth Sciences Technology and Research (GESTAR)
NASA Goddard Space Flight Center
Mail 610.1 | Greenbelt, MD 20771 | USA

Phone: +1 (301) 286-7870
Fax:   +1 (301) 614-6246
Email: abhishek.chatterjee@nasa.gov
Url: https://sciences.gsfc.nasa.gov/sed/bio/abhishek.chatterjee

---

## Author Comment (AC1) · 30 Oct 2020

We thank the reviewers for their detailed suggestions and comments on the manuscript. We have re-written the manuscript, added substantial new analysis, and included extensive new comparisons against independent observations based upon the reviewer suggestions. Below, we have replied to each review and have detailed the corresponding edits that we have made to the manuscript. We have listed out the reviewer comments in italic font and the replies in regular font.

**RC1: Referee #1**

*The authors have developed a geostatistical inverse method to interpret satellite observations of carbon dioxide ($CO_2$) collected by the NASA Orbiting Carbon Observatory collected during 2016. As far as this reviewer can see the study is scientifically sound but describes only an incremental improvement to the method and does not lead to any new scientific insight.*

We have re-written most of the manuscript, overhauled the inverse modeling setup, and added substantial new analysis to improve the novelty and scientific messaging. Specifically, in the revised manuscript, we have added the following new analyses:

- We compare the environmental relationships that we infer from OCO-2 against the relationships that we infer from 15 terrestrial biosphere models (TBMs) from the recent TRENDY model comparison project (Sect. 3.3).
- We evaluate when and where TBMs agree and disagree on these relationships and what factors might be driving these disagreements among TBMs (Sect. 3.1).
- We have expanded the analysis from one year to four years.
- We have added a synthetic data analysis to better explore what factors limit our ability to infer these environmental relationships using current satellite observations from OCO-2 (Sect. 3.1).
- We have added extensive evaluation against ground-based $CO_2$ observations (the Supplemental Sect. S4, Figs. S2-S8, and Tables S2-S3).

*The environmental drivers for ecosystems located at mid/high and tropical ecosystems are unsurprising. Perhaps that's the point. I wasn't sure. PAR is by definition photosynthetic active radiation so its ability to describe large-scale $CO_2$ fluxes isn't anything new, particularly over one year that is dominated by the seasonal cycle. Any insights from using the diffuse and direct components of PAR? Similarly, temperature and precipitation roles in the tropics are nothing new. However, I am surprised that precipitation is such a useful driver over the tropics where complex basin-scale hydrologic controls are at play. In other words, where it rains is not necessary where the water ends up.*

We agree that these environmental drivers are unsurprising. In the revised manuscript, we have added analysis comparing the relationships that we infer from OCO-2 observations against those inferred from 15 TBMs (Sect. 3.3). Existing terrestrial biosphere models (TBMs) disagree on the relationships between these environmental drivers and $CO_2$ fluxes; TBMs show a large range of relationships, and for some variables like precipitation, TBMs often disagree on the sign of that relationship. We feel that this new comparison with process-based models provides better depth and novelty to the manuscript.

It is true that where it rains is not necessarily where water ends up, particularly at fine spatial scales like the scale of a stream catchment. In this study, we model fluxes at a much broader spatial resolution that reflects the resolution of the GEOS-Chem model (4 degrees latitude by 5 degrees longitude). At that broad scale, patterns in spatially-averaged precipitation are more strongly correlated with surface soil moisture than at finer spatial scales. Note that we ran several test simulations where we offered up both precipitation and soil moisture as auxiliary variables in the inverse model, but the model selection framework only chose one of the two (precipitation); those two predictor variables were highly colinear or correlated, indicating that the inverse model did not have the power to distinguish between the two. Furthermore, precipitation was included as a standardized input variable in the TRENDY model inter-comparison, so we wanted to at least offer up precipitation as a candidate auxiliary variable in the analysis of OCO-2 and the TRENDY models.

*The authors have gone some way to 'fess up that the geostatistical inverse method uses prior information for which I commend them. It might not be defined in the same way as the classical Bayesian approach but nonetheless it uses prior information. Otherwise, inferring fluxes for 10ˆ6 grid boxes using 10ˆ5 measurements is an ill-posed problem. The method uses environment driver data with uncertainties that are difficult to quantify (see comment below about estimated posterior uncertainties).*

A geostatistical inverse model certainly does use prior information. That information is just in a different form than other types of Bayesian inverse modeling.

*It would be useful to reiterate to the reader the benefit of the geostatistical inverse method over more traditional methods. Certainly, it provides an alternative perspective but I have seen no evidence to suggest it is better or worse.*

In the revised manuscript, we have added substantial new analysis to better highlight new insights facilitated by this approach. This new analysis includes a comparison of the environmental relationships that we infer from OCO-2 against those inferred from 15 state-of-the-art TBMs (Sect. 3.3). Existing studies have used this geostatistical approach to compare the environmental relationships in different TBMs (e.g., *Huntzinger et al.* 2011) and to compare with the relationships inferred from in situ atmospheric observations (e.g., *Fang and Michalak*, 2015). In the revised study, we build upon that existing body of work by comparing the relationships inferred from OCO-2 across the globe with those inferred from TBMs.

*Line 216: This reader is surprised that OCO-2 data are not sensitive to biomass burning emissions, particularly during the El Nino period. The manuscript would benefit from having more explanation on this point.*

We have overhauled the inverse modeling setup to include more prior information on biomass burning (from GFED) and ocean fluxes (Sect. 2.4). We have also added a new discussion in the results (Sect. 2.4) and SI (the Supplemental Sect. S2) describing the contribution of biomass burning fluxes relative to other types of fluxes. In these sections, we also discuss why biomass burning fluxes are challenging to uniquely identify and constrain in an inverse model. Specifically, the atmospheric signal from biomass burning (as estimated by GFED) is small (0.19

ppm) relative to anthropogenic emissions (2.7 ppm) and model-data errors specified in the inverse model (standard deviation of 0.29 ppm to 4.8 ppm).

*Why are correlations higher when environmental drivers are passed through the atmospheric model. Figure 3 doesn't cut it - the color scale is almost binary as currently defined. Using the square of the correlation might be a better way to illustrate these calculations.*

We have substantially changed this analysis in the revised manuscript and have instead included a synthetic data study to explore what factors limit our ability to infer these environmental relationships using observations from OCO-2. We agree that Fig. 3 in the original manuscript was confusing and have re-designed this analysis to more clearly communicate the message we intended to communicate. We have cut Figure 3 and have replaced it with new results from the synthetic data study (Sect. 3.1 and Fig. 2 in the revised manuscript).

*Line 263: widespread and prolonged drought conditions, together with large-scale land-use change, is a more accurate description of what's going on over these regions.*

Noted. We have edited Sect. 3.3 in the revised manuscript accordingly.

*Paragraph 298: comparison of the reported work and other groups is weak. Not many people have used v9 of OCO-2 data so I think it would be useful for the readership to provide a more detailed assessment of results compared with past estimates using v7 data. The comparison between the model and independent measurements is minimal (in supplementary information). The uncertainties associated with the posterior estimates are unrealistically small. The classical Bayesian inversion as typically employed underestimates posterior uncertainties so certainly the uncertainties estimates reported with the geostatistical method are grossly underestimated. This reviewer is left wondering why this might be so and how a possible explanatory imbalance between prior and observation uncertainties would influence model selection and the analysis that follows.*

We have added substantial comparisons against independent measurements in the revised manuscript, including comparisons with observations from twenty regular aircraft sites (the Supplemental Sect. S4; Figs. S3-S6), the Atmospheric Tomography Mission (ATom) (Fig. S7), and 18 sites from the Total Carbon Column Observing Network (TCCON) (Fig. S8). Note that, in the revised manuscript, we have used version 9 for the analysis because version 7 observations are now several years outdated and contain much larger observational errors.

We have also compared our results against the most recent provisional results from the inverse modeling inter-comparison (MIP) project that uses version 9 of OCO-2 retrievals (refer to the figure below). We find that our flux estimate is usually close to the ensemble mean of the v9 MIP and is always within one standard deviation of the MIP estimates. Note that the results shown below from the v9 MIP are from the MIP website (*https://www.esrl.noaa.gov/gmd/ccgg/OCO2_v9mip/*) and are provisional results that have not yet been finalized.

[Figure]

**Figure R1**. Comparison of biospheric flux estimates by TransCom region from this study (red) and the v9 MIP (blue). Error bars in the MIP results indicate one standard deviation of flux estimates across the ensemble. Our best estimate is usually close to the mean of the v9 MIP study and is always within one standard deviation of the MIP results. Furthermore, the GIM estimate does not show any consistent bias relative to the MIP ensemble mean.

We have also overhauled the inverse modeling setup and have set improved values for the covariance matrices in the inverse model (**R** and **Q**) (Sect. 2.5 and the Supplemental Sect. S1). For example, the model-data mismatch errors are now based upon the reported errors in the 10-second average OCO-2 data product. We have further estimated the relationships between $CO_2$ fluxes and environmental driver datasets using two different meteorological products (MERRA-2 and CRUJRA) to explore the sensitivity of these results to the choice of meteorology used for the driver datasets. We believe that this revision has yielded better uncertainty estimates in the revised manuscript.

*Sure, the tropical flux estimates are important to discuss. However, are the reviewers are in a position to dismiss the results over tropical North Africa without further explanation. Why did they find themselves in terms of environmental drivers? Surely, their results over tropical Africa aren't exclusively determined by measurements collected over tropical Africa? Do they find that seasonal differences in measurement over tropical Africa lead to a bias in the flux? Answers to these questions would represent a useful contribution to the field.*

*There is almost nothing in the manuscript about the large differences between other geographical areas where we would expect much better agreement, e.g temperature North America, Europe, Eurasian temperate. Without a more comprehensive evaluation of the fluxes it is difficult to know whether the method is at fault or the data they have used. This manuscript would benefit greatly from a better evaluation of the posterior fluxes.*

We agree that comparing our results using version 9 of the observations against studies that used version 7 is not necessarily a fair comparison; there are large differences between v7 and v9 of the OCO-2 observations, and differences between existing studies using version 7 and our results using version 9 could reflect differences in the observations as much as differences in inverse modeling methodology. When we compare the GIM flux estimate against provisional results from the most recent MIP, we find much better agreement between our results and the MIP; our estimate is always within one standard deviation of the MIP ensemble mean.

In the revised manuscript, we also evaluate our inverse modeling results using numerous ground-based datasets (the Supplemental Sect. S4, Figs. S2-S8, and Tables S2-S3). We feel that these new model-data comparisons provide a much-improved evaluation of the posterior fluxes.

*Line 59: it would be fairer that Chevallier 2018 argues not suggests.*

We have edited this line accordingly.

*For context, it would be useful for the reader to understand that 2016 was an El Nino Year.*

We include four years of observations in the revised manuscript (instead of the one year in the original manuscript). We also point out in Sect. 3.3 that 2015-2016 are El Nino years.

*Line 91: how did the authors decide that four months was a sufficient spin-up period?*

We have clarified this point in the revised manuscript (the Supplemental Sect. S1). We used this setup for the model spin-up because it is the same setup used in *Miller et al*. (2018). We first created an initial condition for 1 Sept., 2012 based on NOAA's Carbon Tracker (CT) product, and used $CO_2$ fluxes from CT to run GEOS-Chem forward for two years until 1 Sept., 2014 when the inverse modeling begins; we ran the CT fluxes through GEOS-Chem for two years to make sure the $CO_2$ mixing ratios are consistent with the GEOS-Chem model grid, and therefore to minimize potential spin-up artifacts due to model transport. We then run the inverse model starting from 1 Sept., 2014, but we consider the result from 2014 as part of an initial model spin-up period and do not use it for analysis.

**SC1: Brad Weir**

*In this work, the authors use a geostatistical inverse modelling approach to infer surface fluxes from observations of column $CO_2$ by the Orbiting Carbon Observatory 2 (OCO-2). Using these estimates, the authors make claims about the environmental drivers of the spatiotemporal variability of surface fluxes. However, their evaluation against independent data (sometimes coarsely defined as "validation") is not sufficient to support these claims.*

We have added extensive evaluation against ground-based observations, including from 20 regular aircraft sites (the Supplemental Sect. S4; Figs. S3-S6), the Atmospheric Tomography Mission (ATom) (Fig. S7), and 18 sites from the Total Carbon Column Observing Network (TCCON) (Fig. S8). We find that model-data biases are small across most of the globe (except at sites near urban regions) and that the standard deviation of the model-data residuals is within the uncertainties specified in the inverse model (i.e., is within the model-data mismatch specified within the covariance matrix). The Supplemental Sect. S4 of the revised manuscript includes a detailed discussion of these model-data comparisons.

We have also compared our flux estimate against provisional results from the most recent OCO-2model inter-comparison (MIP) project, and our flux estimate is typically close to the ensemble mean and always within one standard deviation of the mean (refer to Fig. R1 above).

*Inferring surface carbon fluxes from observations of atmospheric $CO_2$ is an inherently ill-defined problem. Its solution, in any form, requires a number of assumptions that are often poorly constrained by existing scientific knowledge. The authors do a commendable job of explaining that despite erroneous claims in the existing literature to the contrary, geostatistical inverse models do in fact use prior information, just in a different form than more common approaches. What the authors fail to do is support that their surface flux estimates are fit for the scientific purpose at hand. Typically, this is accomplished through comparisons to other independent data products. While pedantic, it seems more and more necessary that we remind ourselves that inferred surface fluxes fall into the prediction step of the Scientific Method. Between that and the analysis step, is the all important testing step. The testing step cannot be shortcut – it is the only thing separating science from plausible guesswork.*

*In order to make claims about the spatiotemporal variability of surface fluxes, the authors must first evaluate the fidelity of their surface fluxes' spatiotemporal variability. While this reviewer admits that there is no ideal method of evaluating global surface fluxes of $CO_2$ on horizontal scales greater than a few tens of kilometers, a greater effort must be made to demonstrate the product is appropriate for the analysis in the text. In particular, the only evaluation of their surface fluxes is that of long-term time mean regional budgets (Figure 6) and simulated $CO_2$ at just a handful of aircraft profiling sites (Figures S5 and S6). If one is to make claims about seasonal cycles, for example, then the seasonal cycle of the inferred fluxes must be evaluated as well. Given the assumptions necessary to make these inferences, it is entirely possible that their long-term time mean budgets are reasonable and their seasonal cycles are not. This is especially important given the documented impact (Basu et al., 2013, ACP; Crowell et al., 2019) that very small seasonal and regional biases from satellite retrievals can have on inferred fluxes. Unless the authors are able to demonstrate the skill of their product in reproducing variations over the*

*same spatiotemporal scales as the scientific analysis, this review does not see how their claims can be supported.*

Thank you for the suggestions. We have greatly expanded the model-data evaluation in the manuscript (the Supplemental Sect. S4; Figs. S2-S8; and Tables S2-S3). In the original manuscript, we compared against a handful of aircraft sites, as the reviewer points out. In the revised manuscript, we compare against numerous additional aircraft sites, as well as comparisons against TCCON, and comparisons against campaign data from ATom.

**RC2: Julia Marshall**
*At first glance it seems that the results of this study make sense, and are consistent with our general understanding of what drives carbon fluxes, with uptake at higher latitudes being mostly radiation-limited while in the tropics there are more complex temperature-precipitation interactions. So far, so good. The paper is well written and clearly structured, making it easy to read. To the careful reader it soon becomes clear that something is going wrong, however, and the limited "validation" and comparison to other results from the literature are insufficient to explain these problems away. While the geostatistical approach is com mendable in that it allows more flexibility in the structure of the prior fluxes, such that perhaps unexpected signals may emerge, it also seems to allow for rather unphysical results, as in this case. Given the fact that the ocean fluxes (a net sink of more than 2 PgC/year) were rejected by the Bayesian Information Criterion (BIC) while the net land fluxes are more or less consistent with other studies, it seems impossible that the global atmospheric growth rate can be matched. It just does not add up.*

We estimated ocean fluxes alongside terrestrial fluxes in the inverse model in the original manuscript but did a poor job of communicating those results. In the revised manuscript, we have not only improved the discussion of ocean fluxes but have also overhauled the inverse modeling setup to include more detailed prior information for ocean fluxes. In the revised manuscript, we use prior information for ocean fluxes from the NASA Estimating the Circulation and Climate of the Ocean (ECCO) Darwin flux product. In our original setup, prior ocean fluxes from Takahashi were not selected using the BIC, and the inverse model instead defaulted to a non-informative prior over the ocean. In the revised setup, we have grouped together ECCO-Darwin, anthropogenic emissions (from ODIAC), and biomass burning emissions (from GFED) into a single column in the auxiliary variable matrix ($\mathbf{X}$). ECCO-Darwin, when included as a separate column of $\mathbf{X}$ is not selected, but a column of $\mathbf{X}$ that includes all of these prior emissions estimates together is selected.

We describe this updated setup in Sect. 2.4 and the Supplemental Sect. S2 of the revised manuscript and show ocean fluxes alongside terrestrial fluxes in the inverse modeling results in Fig. 3.

*This should be obvious when performing validation, but the very little testing of the posterior fluxes, limited to a handful of aircraft measurements far from coasts on a scatter plot averaged (monthly?) by height, hidden in the supplement, makes it hard*

*to tell. The paper states that aircraft profiles near coasts were not used because the coarse model resolution made it hard to represent these data well, but I wonder if the complete absence of ocean fluxes may have also played a role here?*
*Since none of the in-situ sites were used for constraining the fluxes (which seems an odd choice, even if only for comparison's sake), it would be instructive to plot the concentrations resulting from the posterior fluxes at a few sites to see if the curves drift apart over the year as a result of the missing sink. While this might not look too bad in a simulation of only one year, this would soon result in wildly divergent curves. But perhaps over a longer simulation the BIC would then choose to select the ocean fluxes. Still, the decision to blindly allow the model to return what we know is incorrect makes it hard to trust the interpretation of the results. Perhaps Takahashi was not the best ocean prior in this case, especially for an El Niño year, and this played a role: this could be an area for more analysis.*

In the revised manuscript, we include prior information on ocean fluxes from NASA's ECCO-Darwin product instead of from Takahashi. Recent inverse modeling studies using OCO-2 (e.g., *Liu et al.* 2020) have used the ECCO-Darwin product in place of Takahashi. Existing studies have also shown that ECCO-Darwin exhibits broad consistency with surface ocean $pCO_2$ observations (e.g., *Carroll et al.*, 2020), and the global ocean sink from ECCO-Darwin shows better agreement with the Global Carbon Project (GCP; *Friedlingstein et al.*, 2019) than from Takahashi. We have also added extensive additional model-data comparisons using numerous ground-based datasets. These datasets include 20 regular aircraft sites (the Supplemental Sect. S4; Figs. S3-S6), the Atmospheric Tomography Mission (ATom) (Fig. S7), and 18 sites from the Total Carbon Column Observing Network (TCCON) (Fig. S8). We have further evaluated our flux estimate against provisional results from the most recent OCO-2 model-intercomparison (MIP) (shown in Fig. R1 above).

*The comparison to other model output was largely limited to the OCO-2 model intercomparison study of Crowell et al. (2019), without following the considerable effort they put into validation or consideration of in-situ measurements. Looking at TCCON sites is an obvious choice, as is the extension to additional aircraft measurements, such as AToM, which are available for at least a couple months of 2016. But comparing your (unclosed) budget to the land biosphere budget of other (mass-conserving) studies is intrinsically misleading. (I am not as surprised that BIC did not pick out the GFED emissions, as these are a few orders of magnitude smaller and are easily swallowed up in the biosphere signal.)*

We have included model-data comparisons against both TCCON and ATom in the revised manuscript (the Supplemental Sect. S4 and Figs. S7-S8).

We have also overhauled the inverse modeling setup, and we have reformulated the **X** matrix in the inverse model in a way that ensures the inclusion of more detailed prior information on biomass burning fluxes (We specifically do so by grouping GFED in the same column of **X** with anthropogenic emissions and ocean fluxes.)

*L10 & L204-205: While the difference in wording is subtle, I think the abstract overstates what the meteorological variables explain. Do they really describe 90% of the*

*variability in the fluxes (as seen through OCO-2 observations)? This sort of implies that OCO-2 can "see" fluxes, which isn't true of course. The latter explanation that the deterministic model accounts for XX% of the variance in the estimated fluxes seems more accurate. As you're only treating fluxes on a daily time scale, you're definitely not describing 90% of the variability in the fluxes themselves.*

Thank you for this suggestion. We have revised the wording of the manuscript accordingly.

*Figure 3 and discussion around L235: This is actually quite interesting! I would be interested in seeing some more analysis of this point. It was also not entirely clear to me what was correlated (and how) in Figure 3. The meteorological variables have been "passed through an atmospheric [transport] model": were they then sampled as column-averaged variables, as OCO-2 views the atmosphere? Were the same averaging kernels applied? It also says that this is the correlation "within different global biomes". Were these columns averaged across space then, and the correlation taken in time? Or is this a spatial correlation coefficient between the column-averaged maps for a given time? I feel like there is an intriguing result here, but I don't fully understand what you've done.*

We have added an entire section to the results and discussion to elaborate on this point (Sect. 3.1). We have also revised the analysis described above and instead use synthetic data simulations to better communicate the overarching message of this discussion.

*L238 & 239: How can you be sure that this collinearity is playing a bigger role than retrieval or model errors? Would the latter two effects not also limit the model selection?*

We have added new synthetic data simulations to the manuscript (described in Sect. 3.1) to better explore these questions. In these synthetic simulations, we apply model selection to the original, gridded fluxes from several terrestrial biosphere models (TBMs). We then coarsen the grid of those models to match that of GEOS-Chem and re-apply model selection. In a third case study, we create synthetic OCO-2 observations using those TBMs and apply model selection again, and in the fourth case study, we add estimated model-data errors to those synthetic observations. The case studies make it easier to explore which factors limit our ability to infer relationships between $CO_2$ fluxes and environmental driver variables using current satellite observations from OCO-2.

*L244 & L260: These statements seem to contradict each other. The first says that the negative beta values for PAR mean that an increase in PAR leads to a decrease in NEE and an increase in uptake. The latter says that the negative beta value for scaled temperature means that an increase in temperature leads to reduced uptake. How can these both be true? This is fundamental to the conclusions drawn.*

We have clarified this point in the revised manuscript (Sect. 3.2). An increase in PAR is associated with greater $CO_2$ uptake by the biosphere (i.e., negative NEE). The scaled temperature function is an upside-down parabola, not a monotonically increasing function. At temperatures below 20 – 25 degrees Celsius, an increase in temperature is associated with negative change in

NEE in the inverse model. At temperatures above 20-25 degrees C, an increase in temperature is associated with a positive change in NEE. We also describe this scaled temperature function in detail in the Supplemental Sect. S3 and Fig. S1.

*L257-258: While cloudiness is correlated with clouds and rainfall, it's also correlated with the presence or absence of satellite measurements. What impact might this have on your results?*

Data sparsity in cloudy regions is certainly an issue for satellite-based greenhouse gas sensors. This issue likely increases the uncertainty in our estimated coefficient for precipitation, particularly in wet climates like tropical forests. It may also be one factor in why we only selected a limited number of environmental driver datasets in many biomes. We point out and discuss this issue in Sect. 3.3 of the revised manuscript.

*L302: I'm actually surprised Australia matches as well as it does, as you've had to fold the Southern Ocean sink into the Southern Hemisphere land fluxes somehow.*

We did not do a good job of describing the treatment of ocean fluxes in the inverse model. We have both improved the description of ocean fluxes and have overhauled the inverse modeling setup to more explicitly include a prior ocean flux estimate within the inverse model.

**RC3: Abhishek Chatterjee**

*This begs the question – is this study intended to demonstrate that the GIM approach has been successfully adapted to remote-sensing observations (i.e., a technical study) or is it intended to capture the connections between $CO_2$ fluxes and environmental drivers (i.e., a scientific study)? Kindly see Major Comment #1.*
    *I believe the authors ideally wanted it to address a bit of both but unfortunately, in trying to address both, the authors end up addressing neither. I highly recommend that the authors take a step back and decide whether to focus on the inversion methodology and application to OCO-2 retrievals OR highlight the scientific questions related to regional and seasonal environmental drivers, and then resubmit. In general, the manuscript is well-written and concise, but it falls short of a clear formulation in terms of scientific scope, depth and novelty.*

We have re-written the manuscript and focused on the second question described above (the connections between $CO_2$ fluxes and environmental drivers). We have also de-emphasized the technical or methodological components. We hope that the revised manuscript has a much clearer formulation in terms of scope, depth, and novelty.

*Several other questions persist. These revolve around limited validation of the posterior flux estimates or posterior $CO_2$ concentrations (see Major Comment #4). The choice of the model-data-mismatch variance (R) is inconsistent with real OCO-2 retrievals and needs justification in the main text (rather than bypassing it and relegating it to the Supplementary Section). R, along with the a priori flux covariance matrix Q, balances the relative weight of the*

*atmospheric data and the trend in estimating the fluxes. An inverse modeling study cannot gloss over these details (see Major Comment #6).*

We have greatly expanded model evaluation and have overhauled the inverse modeling setup, including the model-data-mismatch variance. These points are discussed in greater detail below in reply to individual reviewer comments.

*Scope of the study – as mentioned earlier, the authors need to lay out a clear scope early on and remain consistent throughout. If the authors are interested in examining the relationship between carbon flux and environmental drivers, a one-year study is not justifiable. The authors need to examine the relationship over a number of years, make sure they are capturing the inter-annual variability in their flux estimates and then assess the relationship between drivers and fluxes. In addition, it is worth noting that the selected year is an El Niño year. On Page 3, Lines 86 – 88, the authors justify this decision by pointing out that the OCO-2 observations had 7-week gap in 2015- and 1.5-month gap in 2017. Remote sensing datasets, or rather any real observations, will always have data gaps! Simply discarding entire years' worth of data for a 5-7-week gap is not a reasonable justification. On the other hand, if the authors want to highlight the development of a new inversion framework/methodology, then it may be out of scope for ACP, and may be better suited to a journal like GMD, where a lot of the mathematical nuances can be captured. Right now, a lot of the important mathematical details have been relegated to the supplemental material, including important discussions about the error covariance parameters and how they are derived. These details need to be included in the main text.*

We have re-written the manuscript to focus on the relationships between carbon fluxes and environmental drivers and have de-emphasized the inversion framework or methodology. We feel that these environmental relationships make for a more interesting scientific study than focusing on methodological questions, and we hope that this re-write has yielded a manuscript with a much clearer purpose and scope. As part of this revision, we have expanded the time period of the study from one year (2016) to four years (2015 - 2018). In addition, we have included extensive comparisons with terrestrial biosphere models (TBMs) to improve the depth and novelty of the analysis in the manuscript (Sects. 3.1 and 3.3). Specifically, in the revised manuscript, we compare the environmental relationships that we infer from OCO-2 with the environmental relationships that we infer from 15 state-of-the-art TBMs from the recent TRENDY model comparison project.

*Scientific novelty – The authors report that a combination of PAR, daily temperature and daily precipitation are the most adept at capturing the variability in the fluxes (PAR for midto-high latitudes and a combination of daily temperature and precipitation for the tropical biomes). Neither of these findings are unique. The authors have correctly referred to a host of studies using GIM (e.g., Gourdji et al. 2008, Fang and Michalak, 2015, among others) or studies using OCO-2 data that have examined the response of the land carbon cycle during the 2015-2016 El Niño (e.g., Liu et al., 2017, Crowell et al., 2019). The BIC did its job and picked up the variables it was supposed to; hence, it is slightly unclear how this study adds new insights into our knowledge about carbon cycle science. In fact, by the authors own admission in Sections 3.1.1 and 3.1.2, almost all their findings are exactly the same as*

*reported in previous studies. These two sections almost read like a literature review rather than a results section with new and exciting science results.*

We have added substantial new analysis to the revised manuscript to improve the novelty and depth of the scientific results. Specifically, we not only infer environmental relationships using observations from OCO-2 but also compare those against the environmental relationships inferred from 15 TBMs for the same time period. Using OCO-2, we find stronger relationships between temperature and $CO_2$ fluxes across tropical biomes compared to many TBMs, and we find that increases in precipitation across the tropics are associated with greater carbon uptake across seasonal time scales and biome-level spatial scales, a result that disagrees with about half of the TBMs that estimate the opposite relationship. Overall, there are large uncertainties in the environmental relationships within TBMs across all global biomes. The relationships with precipitation are most uncertain in these models while TBMs show greatest agreement on the relationships with temperature. This disagreement over the relationship with precipitation may be due, at least in part, to large disagreements over the fate of precipitation in these ecosystems; each the TRENDY models input the same precipitation estimate but yield evapotranspiration that differs by up to a factor of three among models, depending upon the season and biome. The revised manuscript highlights both the opportunities for informing TBM development using atmospheric observations but also the challenges of doing so using current satellite-based datasets of $CO_2$.

*Selection of auxiliary variables and how they are being reported – what may add a new dimension, relative to already published studies, is reporting a table with all the 12 selected environmental drivers and including the estimated drift coefficients, coefficient of variation, annual average contribution to flux and the correlation coefficient between the selected auxiliary variables in the model of the trend. Actually, the annually averaged global contribution to flux can be reported in typical carbon flux units (like GtC/yr or PgC/yr). That would be novel information, especially if it were to be compared against estimates based on in situ data. Finally, just out of curiosity, why didn't the authors select fPAR instead of PAR? Also, the authors argument for not including LAI or SIF because they are "remote sensing indices" (Page 5, Lines 144-146) is surprising. Almost all of the auxiliary variables listed on Lines 138-141 are derived from remote-sensing measurements. What if the authors were to include LAI? How would that change their selected model of the trend?*

We have greatly expanded the discussion of the auxiliary variables in the re-written manuscript. For example, we have included scatter plots showing the estimated coefficients for each year (Fig. 5), compared those coefficients against coefficients estimated from 15 TBMs (Fig. 4a), and showed the coefficient of variation (as suggested by the reviewer, Fig. 4b). All of the coefficients in the manuscript are listed in units of flux ($\mu$mol m$^{-2}$ s$^{-1}$), so we can better compare the coefficients among different auxiliary variables and different biomes.

Note that in the revised manuscript, we have included PAR instead of fPAR. This was an oversight on our behalf. Furthermore, we decided not to include remote sensing indices in this manuscript because we wanted to focus on comparing the environmental processes in state-of-the-art TBMs against the relationships that we infer from OCO-2. Some TBMs use remote sensing indices like SIF, but some do not. Hence, we felt that it was more appropriate to focus on

environmental processes instead of vegetation indices like SIF or LAI that may not be applicable to many of the TBMs compared in the manuscript. Hence, all of the auxiliary variables used in the revised manuscript are from meteorological reanalysis. We wanted to clearly focus the scope of this manuscript on environmental processes, but we think that an examination of remote sensing indices and global carbon fluxes would make for an interesting future study.

*More rigorous evaluation of posterior flux estimates and more importantly, posterior concentrations, against independent measurements – The biggest surprise of this study is that there are extremely limited evaluations presented against independent measurements (only 7 aircraft sites!). Given the large number of available independent datasets (in situ such as surface flask sites, towers and aircraft, TCCON), the absence of a detailed evaluation is striking. Especially, from a seasoned inverse modeling team. Since the authors claim that they are estimating daily global CO$_2$ fluxes at the GEOS-Chem grid scale (Page 3, Lines 72-73), there should be no reason for not evaluating against observations from dedicated aircraft campaigns such as ATom or ACT-America. In addition, it is also not clear why in Section S7, the authors allude to the results from Crowell et al. 2019. The authors have to back up their own biases and RMSD and explain those numbers and their significance, rather than pointing the reader to Crowell et al. 2019 for justification.*

We have greatly expanded model-data comparisons in the revised manuscript. In the new manuscript, we evaluate the model-data residuals both for the full posterior flux estimate and for the component of the fluxes that is described by the auxiliary variables. In addition, we compare against numerous independent datasets, including 20 regular aircraft sites (the Supplemental Sect. S4; Figs. S3-S6), the Atmospheric Tomography Mission (ATom) (Fig. S7), and 18 sites from the Total Carbon Column Observing Network (TCCON) (Fig. S8). We also provide model-data evaluations for each year of the four-year study period to show that there is no trend in the model-data comparisons (Fig. S2).

*Comparison of findings against those derived from in situ data – The value of this study will be significantly enhanced, if the authors do the same analyses utilizing in situ data (such as NOAA obspack). Are the conclusions, especially in terms of the three significant drivers and their contribution to the carbon flux, consistent? It has been 12+ years since the Gourdji et al. 2008 study attempted such an analysis – given the increase in the number of surface flask sites and improvements in atmospheric transport model, availability of auxiliary datasets, it will be worth revisiting this and comparing against the information reported here from OCO$_2$ datasets.*

Several of the reviewers, including this reviewer, recommended defining a more targeted scope and more clearly defined aims in the manuscript, and we have tried to do so in the re-written manuscript. The focus of this manuscript is estimating the relationships between CO$_2$ fluxes and environmental driver datasets using OCO-2 and comparing those inferences against the relationships estimated from 15 state-of-the-art TBMs. We agree that an *in situ* data study would be interesting, but we feel that this focus would be better left for a separate study in the interest of maintaining a targeted scope with clearly defined aims. Furthermore, the results and discussion section of the revised manuscript are heavily focused on the tropics, and the in-situ observation network is very sparse across the tropics; existing studies have raised questions

about the strength of the tropical flux constraint in in-situ inversions (e.g., *Crowell et al*. 2019; *Piao et al*. 2020).

*Error covariance parameters – Can the authors explain why they switched to a spherical covariance model instead of sticking with a simpler exponential covariance model? The authors argue that the shorter correlation length is due to higher density of observations relative to previous studies. Part of that is true. But I believe that the shorter correlation length in the residuals is more reflective of the model of the trend that has been fitted to large biome scales. The model of the trend is too complex for the biome scale; for the grid scale studies that the authors allude to, it made sense. Additionally, the authors persist with a model-data mismatch variance of 1.19 ppm2 based on a previous pseudo-data study. Why? I highly encourage the authors to use the reported XCO$_2$ uncertainty for the OCO-2 soundings and then add reasonable representation of transport and representation errors to get 'real' MDM variances. This shouldn't be a huge task given the involvement of core GEOS-Chem developers in this study. It wouldn't be surprising if more reasonable R values lead to an increase in a posteriori uncertainties for their flux estimates (Page 11, Lines 324-325).*

We overhauled the inverse modeling setup in response to suggestions from reviewers and have changed the covariance matrix parameters in the inverse model as suggested by this reviewer. Specifically, we use estimated model-data mismatch errors from the 10-second OCO-2 data product (e.g., *Crowell et al*. 2019), described in Sects. 2.5 and the Supplemental Sect. S1. In addition, we use an exponential model for the **Q** covariance matrix. Note that a spherical is very similar to an exponential model, but a spherical model decays to zero, unlike an exponential model which decays to near-zero but never actually reaches zero (e.g., *Kitanidis*, 1997). A spherical model therefore yields covariance matrices that require substantially less computer memory, a particular benefit for large inverse problems (e.g., *Miller et al*. 2020). In this study, the components of **Q** are small enough such that we were able to use an exponential model.

**References:**

Carroll, D., Menemenlis, D., Adkins, J.F., Bowman, K.W., Brix, H., Dutkiewicz, S., Fenty, I., Gierach, M.M., Hill, C., Jahn, O. and Landschützer, P.: The ECCO-Darwin Data-Assimilative Global Ocean Biogeochemistry Model: Estimates of Seasonal to Multidecadal Surface Ocean $pCO_2$ and Air-Sea $CO_2$ Flux. Journal of Advances in Modeling Earth Systems, *12*(10), p.e2019MS001888. https://doi.org/10.1029/2019MS001888, 2020.

Crowell, S., Baker, D., Schuh, A., Basu, S., Jacobson, A. R., Chevallier, F., Liu, J., Deng, F., Feng, L., McKain, K., Chatterjee, A., Miller, J. B., Stephens, B. B., Eldering, A., Crisp, D., Schimel, D., Nassar, R., O'Dell, C. W., Oda, T., Sweeney, C., Palmer, P. I., and Jones, D. B. A.: The 2015–2016 carbon cycle as seen from OCO-2 and the global in situ network, Atmos. Chem. Phys., 19, 9797–9831, https://doi.org/10.5194/acp-19-9797-2019, 2019.

Fang, Y., and Michalak, A. M.: Atmospheric observations inform $CO_2$ flux responses to enviroclimatic drivers. Global Biogeochemical Cycles, 29(5), 555-566. https://doi.org/10.1002/2014/GB005034, 2015.

Friedlingstein, P., Jones, M. W., O'Sullivan, M., Andrew, R. M., Hauck, J., Peters, G. P., Peters, W., Pongratz, J., Sitch, S., Le Quéré, C., Bakker, D. C. E., Canadell, J. G., Ciais, P., Jackson, R. B., Anthoni, P., Barbero, L., Bastos, A., Bastrikov, V., Becker, M., Bopp, L., Buitenhuis, E., Chandra, N., Chevallier, F., Chini, L. P., Currie, K. I., Feely, R. A., Gehlen, M., Gilfillan, D., Gkritzalis, T., Goll, D. S., Gruber, N., Gutekunst, S., Harris, I., Haverd, V., Houghton, R. A., Hurtt, G., Ilyina, T., Jain, A. K., Joetzjer, E., Kaplan, J. O., Kato, E., Klein Goldewijk, K., Korsbakken, J. I., Landschützer, P., Lauvset, S. K., Lefèvre, N., Lenton, A., Lienert, S., Lombardozzi, D., Marland, G., McGuire, P. C., Melton, J. R., Metzl, N., Munro, D. R., Nabel, J. E. M. S., Nakaoka, S.-I., Neill, C., Omar, A. M., Ono, T., Peregon, A., Pierrot, D., Poulter, B., Rehder, G., Resplandy, L., Robertson, E., Rödenbeck, C., Séférian, R., Schwinger, J., Smith, N., Tans, P. P., Tian, H., Tilbrook, B., Tubiello, F. N., van der Werf, G. R., Wiltshire, A. J., and Zaehle, S.: Global Carbon Budget 2019, Earth Syst. Sci. Data, 11, 1783–1838, https://doi.org/10.5194/essd-11-1783-2019, 2019.

Huntzinger, D. N., Gourdji, S. M., Mueller, K. L., and Michalak, A. M.: A systematic approach for comparing modeled biospheric carbon fluxes across regional scales, Biogeosciences, 8, 1579–1593, https://doi.org/10.5194/bg-8-1579-2011, 2011.

Kitanidis, P.: Introduction to Geostatistics: Applications in Hydrogeology, Stanford-Cambridge program, Cambridge University Press, Cambridge, 1997.

Liu, J., Baskaran, L., Bowman, K., Schimel, D., Bloom, A. A., Parazoo, N. C., Oda, T., Carroll, D., Menemenlis, D., Joiner, J., Commane, R., Daube, B., Gatii, L. V., McKain, K., Miller, J., Stephens, B. B., Sweeney, C., and Wofsy, S.: Carbon Monitoring System Flux Net Biosphere Exchange 2020 (CMS-Flux NBE 2020), Earth Syst. Sci. Data Discuss., https://doi.org/10.5194/essd-2020-123, in review, 2020.

Miller, S. M., Michalak, A. M., Yadav, V., and Tadić, J. M.: Characterizing biospheric carbon balance using $CO_2$ observations from the OCO-2 satellite, Atmos. Chem. Phys., 18, 6785–6799, https://doi.org/10.5194/acp-18-6785-2018, 2018.

Miller, S. M. and Michalak, A. M.: The impact of improved satellite retrievals on estimates of biospheric carbon balance, Atmos. Chem. Phys., 20, 323–331, https://doi.org/10.5194/acp-20-323-2020, 2020.

Piao, S., Wang, X., Wang, K., Li, X., Bastos, A., Canadell, J. G., ... & Sitch, S.: Interannual variation of terrestrial carbon cycle: Issues and perspectives. Global Change Biology, 26(1), 300-318, 2020.